# Structural basis of RNAPII transcription on the nucleosome containing histone variant H2A.B

Munetaka Akatsu [ID] [1,2], Rina Hirano [ID] [1,2], Tomoya Kujirai[1,3], Mitsuo Ogasawara [ID] [1], Haruhiko Ehara [ID] [3], Shun-ichi Sekine [ID] [3], Yoshimasa Takizawa[1,4] & Hitoshi Kurumizaka [ID] [1,2,3 ✉]

## Abstract

H2A.B is a distant histone H2A variant associated with actively transcribed regions of the genome, suggesting its positive role in promoting transcription. In the present study, we demonstrate that the RNA polymerase II elongation complex (EC) transcribes the nucleosome containing H2A.B more efficiently than that with canonical H2A in vitro. Our cryo-electron microscopy analysis of the H2A.B nucleosome during transcription revealed that the proximal H2A.B-H2B dimer is released from the nucleosome as the EC transcribes the proximal half of the nucleosomal DNA. This dissociation, which is not observed in the canonical H2A nucleosome, likely enhances the EC elongation efficiency in the H2A.B nucleosome. Mutational analyses suggested that the unique short C-terminal region of H2A.B alone enhances EC elongation efficiency when substituted for its counterpart in canonical H2A. Additionally, other regions of H2A.B contribute to this enhancement. These structural and biochemical findings provide new insights into the role of H2A.B in regulating gene expression.

**Keywords** Nucleosome; Chromatin; Transcription; Histone Variant; H2A.B
**Subject Categories** Chromatin, Transcription & Genomics; Structural Biology

## Introduction

The nucleosome is the fundamental repeating unit of chromatin in eukaryotes. In the nucleosome, two histone H2A-H2B dimers and a histone H3-H4 tetramer form the histone octamer, in which 145–147 base pairs of DNA are symmetrically wrapped with approximately ten base-pair periodic histone DNA contacts at superhelical locations (SHLs) −7, −6, −5, −4, −3, −2, −1, 0 (dyad), +1, +2, +3, +4, +5, +6, and +7 (Luger et al, 1997). Since the nucleosome has a stable architecture due to these multiple

histone-DNA contacts, the machinery regulating genomic DNA functions must overcome this nucleosome barrier (Wolffe, 1999).

In the gene expression process, RNA polymerase II (RNAPII) is recruited to the promoter region, which is located upstream of the coding DNA region and usually maintained as a "nucleosome-free" DNA region in eukaryotic genomes (Haberle and Stark, 2018; Girbig et al, 2022). RNAPII proceeds from the promoter to the protein coding regions, in which nucleosomes are aligned and form the beads-on-a-string architecture (Li et al, 2007; Jonkers and Lis, 2015). The nucleosomal DNA is gradually peeled from the histone surface by the progression of RNAPII, which transiently pauses at the nucleosomal SHL(−6), SHL(−5), SHL(−2), and SHL(−1) positions until reaching SHL(0) (Kujirai et al, 2018a). Transcription elongation factors, Spt4/5 (DSIF for mammalian) and Elf1 (ELOF1 for mammalian), drastically reduce these RNAPII pauses within the nucleosome (Ehara et al, 2019). The nucleosome is then disassembled when the RNAPII passes through the SHL(0) position of the nucleosome, and is immediately reassembled on the upstream transcribed DNA region with the aid of the histone chaperone FACT (Ehara et al, 2022).

Canonical histones are newly produced during the S phase of the cell cycle and are incorporated into chromatin (Groth et al, 2007; Marzluff and Duronio, 2002; Marzluff et al, 2008). In contrast, histone variants are produced in a cell-cycle independent manner and encoded by distinct genes (Nekrasov et al, 2012; Boyarchuk et al, 2014; Henikoff and Smith, 2015). Histone variants increase the diversity of nucleosomes in terms of their structure, dynamics, and stability, and are considered to function in genome regulation by chromatin (Suto et al, 2000; Chakravarthy et al, 2005; Tachiwana et al, 2010; Tachiwana et al, 2011; Urahama et al, 2014; Sharma et al, 2019; Kurumizaka et al, 2021).

Among the H2A variants, H2A.B is the most distant, with about 50% amino acid identity and a shorter C-terminal tail compared to canonical H2A (Chadwick and Willard, 2001; Fig. 1A). H2A.B is primarily expressed in testis, and may function in the extensive chromatin reorganization during spermatogenesis (Ishibashi et al, 2010; Soboleva et al, 2011; Hoghoughi et al, 2018). In addition, H2A.B lacks the acidic residues forming the nucleosomal acidic patch, which provides a docking site for nucleosome-binding proteins, such as histone modifiers, histone chaperones, and

[1]Laboratory of Chromatin Structure and Function, Institute for Quantitative Biosciences, The University of Tokyo, 1-1-1 Yayoi, Bunkyo-ku, Tokyo 113-0032, Japan. [2]Department of Biological Sciences, Graduate School of Science, The University of Tokyo, 1-1-1 Yayoi, Bunkyo-ku, Tokyo 113-0032, Japan. [3]Laboratory for Transcription Structural Biology, RIKEN Center for Integrative Medical Sciences, 1-7-22 Suehiro-cho, Tsurumi-ku, Yokohama 230-0045, Japan. [4]Department of Computational Biology and Medical Sciences, Graduate School of Frontier Sciences, The University of Tokyo, 1-1-1 Yayoi, Bunkyo-ku, Tokyo 113-0032, Japan. ✉E-mail: kurumizaka@iqb.u-tokyo.ac.jp

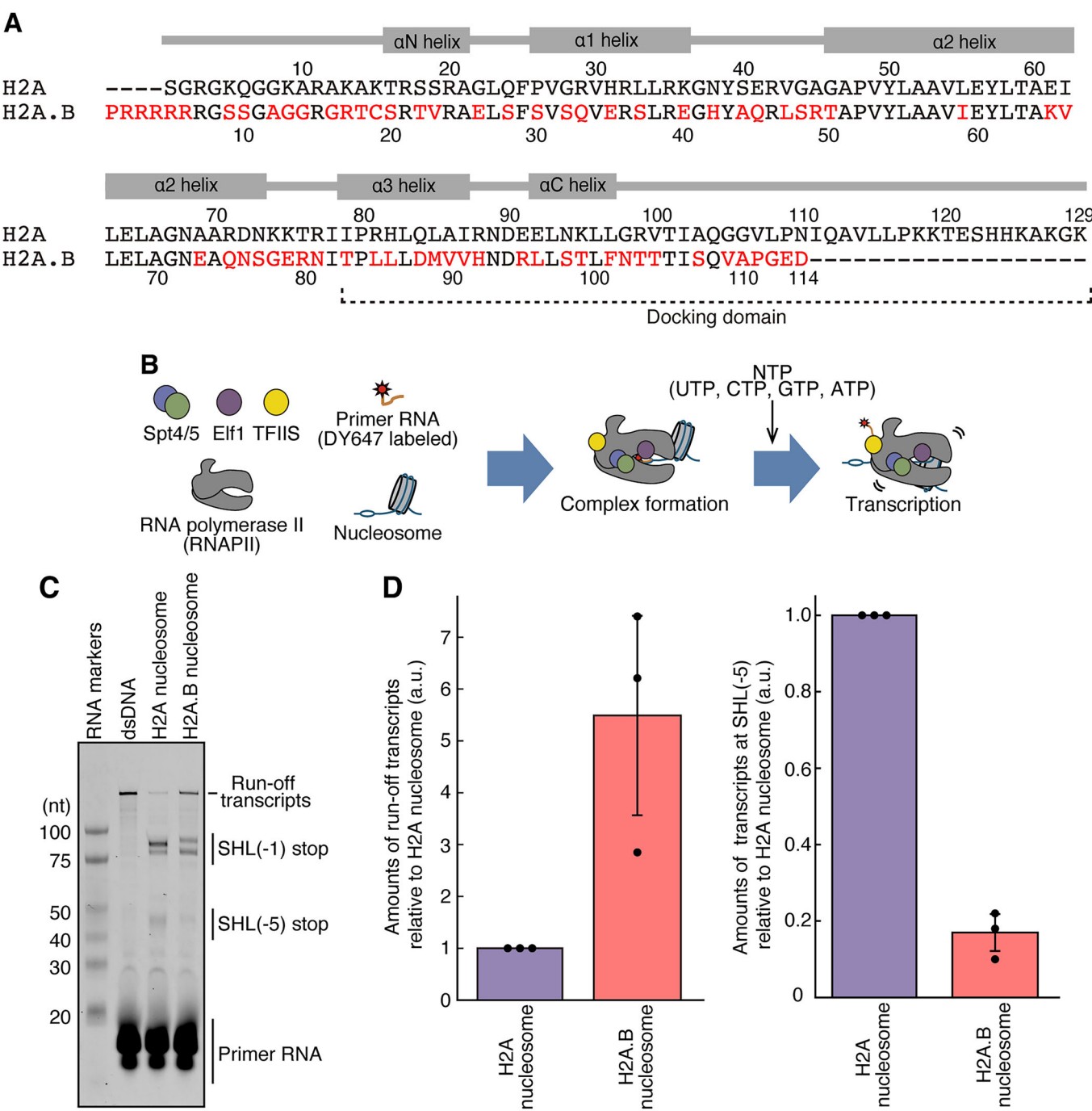

**Figure 1.  RNA polymerase II efficiently transcribes on the H2A.B nucleosome.**

(A) Amino acid sequences of histones H2A and H2A.B. Red characters indicate non-conserved amino acid residues between H2A and H2A.B. (B) Schematic representation of the nucleosome transcription assay by RNAPII with elongation factors, Spt4/5, Elf1, and TFIIS. (C) The nucleosome transcription assay. RNA transcripts were detected with the DY647 fluorescent dye. Reproducibility was confirmed by three independent experiments (shown in Appendix Fig. S2A,B). (D) Quantification of the transcription assay. Band intensities of the run-off transcripts (left panel) and transcripts paused at the SHL($-5$) position (right panel) were quantitated. Amounts of the transcripts in the H2A.B nucleosome were plotted relative to those in the H2A nucleosome (a.u.). Error bars indicate standard deviations ($n = 3$ independent replicates). The central values of the error bars for the run-off and SHL($-5$) transcripts (i.e., the mean values of the plots) are 5.49 and 0.17 for the H2A.B nucleosome, respectively. Source data are available online for this figure.

nucleosome remodelers (Angelov et al, 2004; Zhou et al, 2007; Soboleva et al, 2011). H2A.B transiently accumulates at DNA replication and repair sites (Arimura et al, 2013; Sansoni et al, 2014). In vitro, H2A.B forms a nucleosome with flexible DNA ends, and efficiently exchanges the H2A.B-H2B dimer with the canonical H2A-H2B dimer through an intermediate "open conformation" (Bao et al, 2004; Doyen et al, 2006, Arimura et al, 2013; Sugiyama et al, 2014; Hirano et al, 2021; Zhou et al, 2021; Nozawa et al, 2024). Therefore, H2A.B renders the nucleosomal DNA more accessible than canonical H2A. Interestingly, H2A.B reportedly exists on transcribing genes, including the transcription start sites and/or gene body regions (Ioudinkova et al, 2012; Tolstorukov et al, 2012; Nekrasov et al, 2013; Chen et al, 2014; Soboleva et al, 2017). These facts suggest that H2A.B may facilitate transcription by its incorporation into the nucleosome.

In the present study, we found that the H2A.B nucleosome is transcribed by the RNA polymerase II (RNAPII) elongation complex more efficiently than the canonical H2A nucleosome. The cryo-electron microscopy (cryo-EM) structures of the transcribing RNAPII elongation complex-H2A.B nucleosome complexes explain the mechanism by which H2A.B enhances the efficiency of RNAPII transcription on the nucleosome.

## Results

### RNAPII elongation complex efficiently transcribes the H2A.B nucleosome

To study the effect of the histone H2A.B variant on chromatin transcription, we conducted a nucleosome transcription assay. Purified *Komagataella phaffii* (formerly *K. pastoris* or *Pichia pastoris*) RNAPII, along with TFIIS, Spt4/5, and Elf1, were combined to form the RNAPII elongation complex (EC), which was then used for nucleosome transcription (Fig. 1B; Appendix Fig. S1A–D). Consistent with the previous report (Ehara et al, 2019), the canonical H2A nucleosome exhibited EC pausing at the SHL(−5) and SHL(−1) positions (Fig. 1C; Appendix Fig. S2A,B, lane 3). The formation of the nucleosome also drastically reduced the production of the full-length run-off transcript (Fig. 1C,D; Appendix Fig. S2A,B, lanes 2 and 3). Interestingly, in the H2A.B nucleosome, the EC-mediated production of the run-off transcript was markedly increased, and accompanied by a substantial reduction in pausing at the SHL(−5) position (Fig. 1C,D; Appendix Fig. S2A,B, lane 4). The SHL(−1) pausing of the EC was also reduced in the H2A.B nucleosome at a slightly different pausing position, probably due to its unstable nucleosome positioning (Fig. 1C,D; Appendix Fig. S2A,B, lane 4). These results indicate that EC-mediated transcription occurs more efficiently in the H2A.B nucleosome, compared to the canonical H2A nucleosome.

### Cryo-EM structures of the RNAPII elongation complex transcribing on the H2A.B nucleosome

To reveal the mechanism by which EC traverses the H2A.B nucleosome, we determined the structure of the EC-H2A.B nucleosome complex paused at the SHL(−1) position by cryo-EM single particle analysis. We conducted the transcription reaction by the EC with the H2A.B nucleosome, and the EC was intentionally paused at the SHL(−1) position by performing the transcription reaction in the absence of ATP. Under these conditions, the EC was paused at the position 42 base pairs from the nucleosome entry site (Fig. 2A). We also prepared the EC-H2A.B nucleosome complex paused at the SHL(−5) position by omitting CTP and ATP in the reaction mixture (Fig. 2A). The resulting complexes containing the paused EC, in which the RNAPII leading edges had reached the SHL(−1) and SHL(−5) positions, were fractionated by sucrose density gradient ultracentrifugation with glutaraldehyde (GraFix) (Kastner et al, 2008) (Appendix Fig. S3A–D).

We first obtained the EC-H2A.B nucleosome structure paused at the SHL(−5) position (Figs. 2B, EV1, and EV2). The RNAPII pausing position was deduced from the length of the RNA transcript (Appendix Fig. S3B). In this structure, the resolution of the nucleosome part was relatively low (6.9 Å) due to the flexibility of the entry nucleosomal DNA region in the H2A.B nucleosome. However, we successfully visualized the cryo-EM maps corresponding to the H2A.B-H2B dimers on both sides of the nucleosome (Fig. 2C). This indicates that the proximal H2A.B-H2B dimer remained associated when the EC paused at this position (Fig. 2B,C).

Surprisingly, in the EC-H2A.B nucleosome structure paused at the SHL(−1) position at 4.8 Å resolution, the cryo-EM map corresponding to the proximal H2A.B-H2B dimer was not visualized, suggesting that the H2A.B-H2B dimer proximal to RNAPII is dissociated from the nucleosome (Figs. 2C, right panels, EV3, EV4, and EV5). In the SHL(−1) complex, the RNA nucleotides within the RNAPII catalytic center were visualized, identifying the RNAPII pausing position, where the leading edge of RNAPII was located at the SHL(−1) position (Fig. EV4D). Therefore, the proximal H2A.B-H2B dimer dissociates after RNAPII passes through the SHL(−5) position. In contrast, in the previous cryo-EM structure of the EC-canonical H2A nucleosome complex paused at the SHL(−1) position, two H2A-H2B dimers are maintained together with the H3-H4 tetramer (Ehara et al, 2019) (Fig. 2C, left panel). The proximal H2A.B-H2B dissociation may occur because of the weak association of the H2A.B-H2B dimer with the H3-H4 tetramer in the nucleosome (Bao et al, 2004; Doyen et al, 2006; Hirano et al, 2021; Zhou et al, 2021). The reduced SHL(−5) pausing and enhanced run-off transcription ratios in the H2A.B nucleosome may be consequences of the proximal H2A.B-H2B dissociation during the RNAPII passage.

To test whether EC transcription induces H2A.B-H2B dimer dissociation, we performed a DNase I footprinting analysis on EC-nucleosome complexes paused at the SHL(−1) position (Fig. 3A; Appendix Fig. S4). In the H2A.B nucleosome template, the DNase I accessibility was substantially enhanced at 30, 31, 33, and 39 base pairs from the distal end of the nucleosomal DNA (Fig. 3B; Appendix Fig. S5, red asterisks). These hypersensitive sites in the H2A.B nucleosome were detected in an EC transcription-dependent manner, but were not observed in the canonical H2A nucleosome. These regions are located near the position where the proximal H2A.B-H2B dimer was deposited before transcription initiation (Fig. 3C). Accordingly, proximal H2A.B-H2B dissociation may occur in the H2A.B nucleosome, but not in the canonical H2A nucleosome, when the EC traverses the proximal half of the nucleosomal DNA.

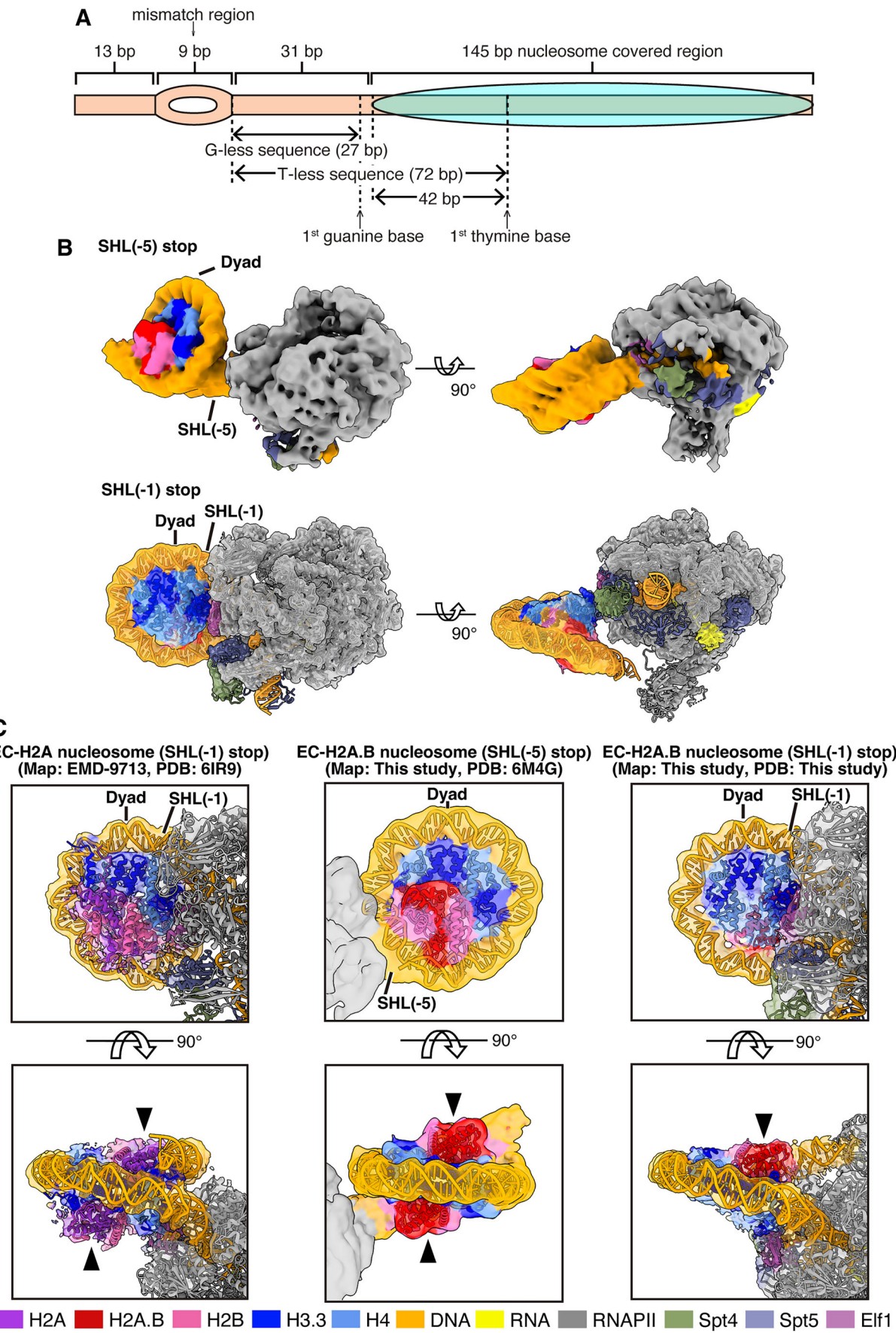

**A**

mismatch region

13 bp | 9 bp | 31 bp | 145 bp nucleosome covered region

G-less sequence (27 bp)

T-less sequence (72 bp)

42 bp

1st guanine base    1st thymine base

**B**

SHL(-5) stop

Dyad

SHL(-5)

90°

SHL(-1) stop

Dyad    SHL(-1)

90°

**C**

EC-H2A nucleosome (SHL(-1) stop)
(Map: EMD-9713, PDB: 6IR9)

Dyad    SHL(-1)

EC-H2A.B nucleosome (SHL(-5) stop)
(Map: This study, PDB: 6M4G)

Dyad

SHL(-5)

EC-H2A.B nucleosome (SHL(-1) stop)
(Map: This study, PDB: This study)

Dyad    SHL(-1)

90°    90°    90°

| H2A | H2A.B | H2B | H3.3 | H4 | DNA | RNA | RNAPII | Spt4 | Spt5 | Elf1 |

◄

**Figure 2.   Cryo-EM structures of Spt4/5-Elf1-RNAPII elongation complexes transcribing the H2A.B nucleosome.**

(**A**) Graphical representation of the nucleosomal DNA template. RNAPII is loaded on the mismatch region with an RNA primer. In the presence of UTP, GTP, and CTP, the RNAPII is paused when the first thymine base is incorporated into its catalytic center, and the leading edge of the RNAPII is located at the SHL(−1) position. In the presence of UTP and GTP, the RNAPII is paused when the first guanine base is incorporated into its catalytic center, and the leading edge of the RNAPII is located at the SHL(−5) position. (**B**) Composite cryo-EM map of the EC-H2A.B nucleosome complexes paused at the SHL(−5) (top panels) and SHL(−1) (bottom panels) positions. Two views are presented. For the SHL(−1) complex, atomic models are represented as a cartoon on a transparent cryo-EM map. (**C**) Close-up views of the nucleosomes in the EC-H2A.B nucleosome complex paused at the SHL(−5) position (center panels, Map: EMD-60952, PDB: 6M4G) and the SHL(−1) position (right panels, Map: EMD-60953, PDB: 9II7). Two views are presented. The atomic model is superimposed on the cryo-EM map. The corresponding region of the EC-canonical H2A nucleosome complex paused at the SHL(−1) position (left panels, Map: EMD-9713, PDB: 6IR9) is presented for comparison. The arrowhead indicates the H2A-H2B dimer or the H2A.B-H2B dimer. Color codes for all panels are shown at the bottom of this figure.

It should be noted that, in DNase I footprinting with the H2A.B nucleosome, DNase I hypersensitive sites were also observed in the 100–140 base-pair DNA region from the distal end of the nucleosomal DNA (Fig. 3B; Appendix Fig. S5, black bar). This region is originally located near the proximal entry DNA region, which is known to be detached from the histone surface in the H2A.B nucleosome (Zhou et al, 2021). This region may be occupied by the RNAPII molecule if it is paused at the designated SHL(−1) position. Therefore, these hypersensitive sites may result from the EC population that approaches or passes the entry region of the H2A.B nucleosome and peels the DNA in front of the EC.

**Mutational analyses of the H2A.B nucleosome**

The amino acid sequence of the H2A.B C-terminal region is not conserved with that of canonical H2A, and its length is shorter (Figs. 1A and 4A). In the previous EC-H2A nucleosome structure paused at the SHL(−5) position, the H2A C-terminal region associates with the N-terminal region of H3 and remains in the nucleosome (Ehara et al, 2019). This H3-H2A association functions to wrap the DNA around the entry/exit regions of the nucleosomal DNA (Tachiwana et al, 2010) (Fig. 4B, left panel). In contrast, in the EC-H2A.B nucleosome structure paused at the SHL(−5) position, the entry/exit region of the nucleosomal DNA is unwrapped due to the lack of DNA binding by the H3 N-terminal region (Zhou et al, 2021) (Fig. 4B, right panel). This may happen because there is no interaction between the H3 N-terminal and H2A C-terminal regions, due to the H2A.B-specific C-terminal composition. This H2A.B-specific feature may enhance the nucleosomal transcription efficiency by the EC.

We then tested the effects of the shortened C-terminal region of H2A.B on the EC transcription efficiency. To do so, we prepared the H2A$^{H2A.B(102-114)}$ mutant, in which the H2A C-terminal region was replaced by residues 102–114 of H2A.B (Hirano et al, 2021) (Fig. 4A; Appendix Fig. S6A,B).

The nucleosome transcription assay with the EC and the H2A$^{H2A.B(102-114)}$ nucleosome revealed that the amount of run-off transcripts was drastically enhanced and the SHL(−5) pausing was significantly reduced, to levels comparable to those with the H2A.B nucleosome (Fig. 4C,D; Appendix Fig. S7A,B). These results suggest that the C-terminal region of H2A.B plays a role in the enhanced nucleosome transcription by the EC.

To investigate the role of other regions of H2A.B in nucleosome transcription, we generated an H2A.B$^{H2A(98-129)}$ mutant, in which the C-terminal region of H2A.B was replaced with residues 98–129 of canonical H2A (Fig. 4A; Appendix Fig. S8A,B). Interestingly, the EC transcription profile of the H2A.B$^{H2A(98-129)}$ nucleosome closely resembled that of the H2A.B nucleosome (Fig. 4C,D; Appendix Fig. S7A,B). These findings suggest that the H2A.B-specific short C-terminal region alone may not be solely responsible for the enhanced EC transcription observed in the H2A.B nucleosome.

# Discussion

H2A.B reportedly accumulates in actively transcribing genes and enhances transcriptional elongation by an unknown mechanism (Ioudinkova et al, 2012; Tolstorukov et al, 2012; Nekrasov et al, 2013; Chen et al, 2014; Soboleva et al, 2017). Consistently, we found that the EC composed of RNAPII, Spt4/5, and Elf1 efficiently transcribes the H2A.B nucleosome, as compared to the canonical H2A nucleosome (Fig. 1). To elucidate the mechanism by which the EC traverses the DNA wrapped in the H2A.B nucleosome, we performed a cryo-EM analysis coupled with a nucleosome transcription reaction. In this method, the EC-nucleosome complexes during the RNAPII progression are captured as snapshot structures (Kujirai et al, 2018a). Our results revealed that the proximal H2A.B-H2B dimer is dissociated from the nucleosome when the EC reaches the SHL(−1) position in the EC-H2A.B nucleosome complex (Figs. 2C and EV5). In contrast, the proximal H2A-H2B dimer is retained when the EC or RNAPII is paused at the SHL(−1) position of the canonical H2A nucleosome (Kujirai et al, 2018a; Ehara et al, 2019; Ehara et al, 2022) (Fig. 2C). Importantly, the proximal H2A.B-H2B dimer is still retained within the H2A.B nucleosome when the EC reaches the SHL(−5) position, indicating that the dissociation of the proximal H2A.B-H2B dimer may be promoted after the transcribing EC passes through the SHL(−5) position of the nucleosome (Fig. 5).

The proximal H2A.B-H2B dimer dissociation could be important for nucleosome reassembly by the EC. When the EC passes through the SHL(0) position, the downstream nucleosome is completely disassembled, and then the upstream nucleosome is immediately reassembled on the DNA behind the EC with the aid of the histone chaperone FACT (Ehara et al, 2022). The proximal H2A-H2B dimer is required for FACT binding, and plays an essential role in the histone transfer and the subsequent nucleosome reassembly. Alternatively, in the absence of histone chaperones/elongation factors, the nucleosome reassembly could potentially be mediated via a template looping intermediate, in which the upstream DNA is rewrapped on the proximal H2A-H2B dimer of the downstream nucleosome (Filipovski et al, 2022; Akatsu et al, 2023). The formation of this intermediate also requires the proximal H2A-H2B dimer. Therefore, the dissociation of the proximal H2A.B-H2B dimer during transcription on the H2A.B

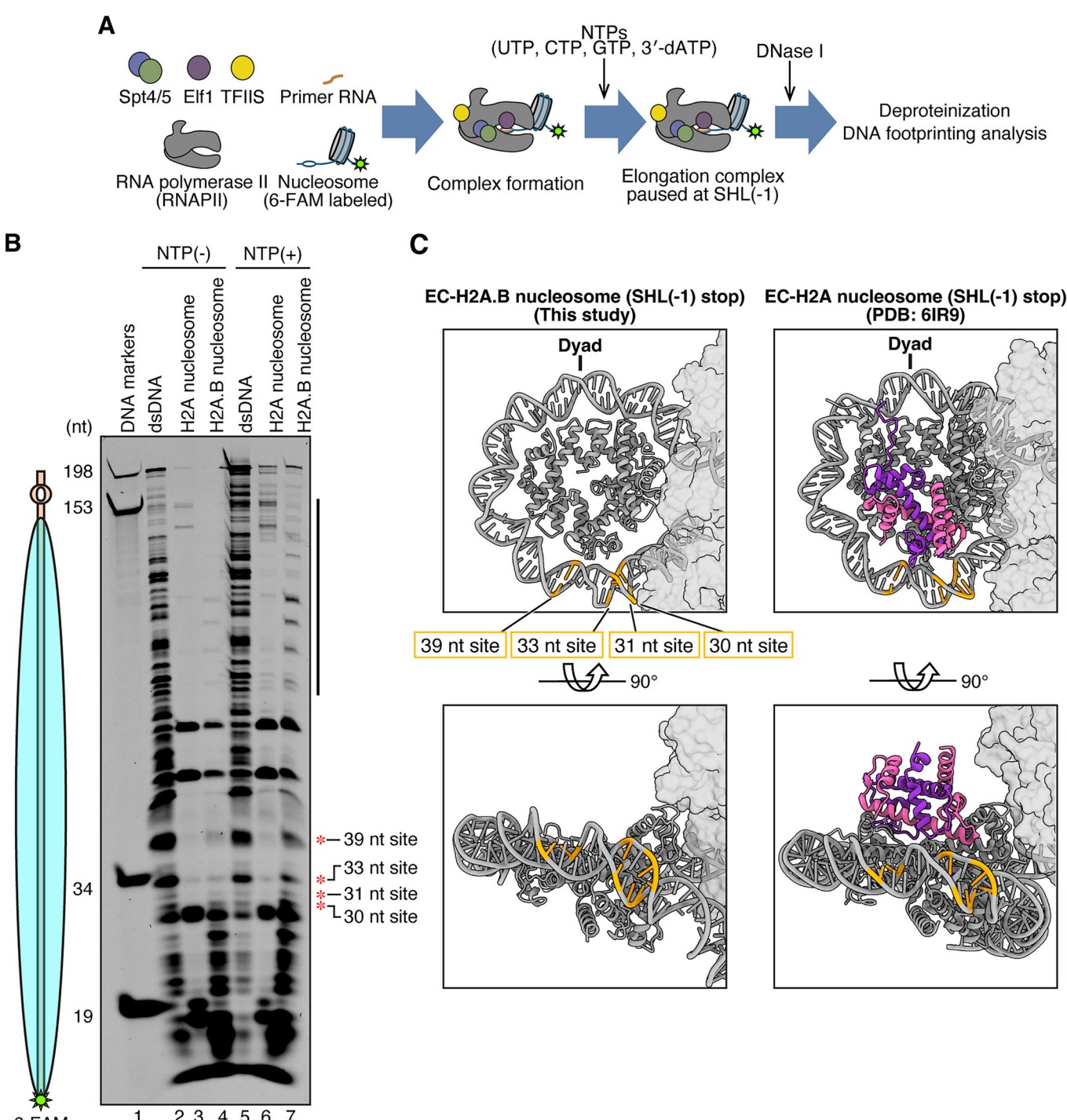

**Figure 3. DNase I footprinting analysis of the Spt4/5-Elf1-RNAPII elongation complexes transcribing on the nucleosome.**

(A) Schematic representation of the DNase I footprinting analysis. The nucleosomal DNA was labeled with 6-FAM fluorescent dye at the opposite end of the transcription start site (bubble DNA). The transcription reaction was conducted in the presence of 3′-dATP, and the EC was paused at the transcription at the SHL(−1) position of the nucleosome. DNase I was then added, and the samples were deproteinized. The resulting DNA fragments were analyzed by denaturing polyacrylamide gel electrophoresis. (B) A representative gel image of the DNase I footprinting analysis. The resulting DNA fragments were detected by the 6-FAM fluorescence signal. Red asterisks and the black bar on the right-hand side of the gel image indicate the DNase I hypersensitive sites induced by the EC transcription in the H2A.B nucleosome. Reproducibility was confirmed by two independent experiments (shown in Appendix Fig. S5). (C) The DNase I hypersensitive sites with red asterisks in (B) are shown on the atomic model of the Spt4/5-Elf1-RNAPII-nucleosome complex. The 39 nt, 33 nt, 31 nt, and 30 nt sites are shown in orange. The proximal H2A and H2B molecules are shown in purple and pink, respectively.

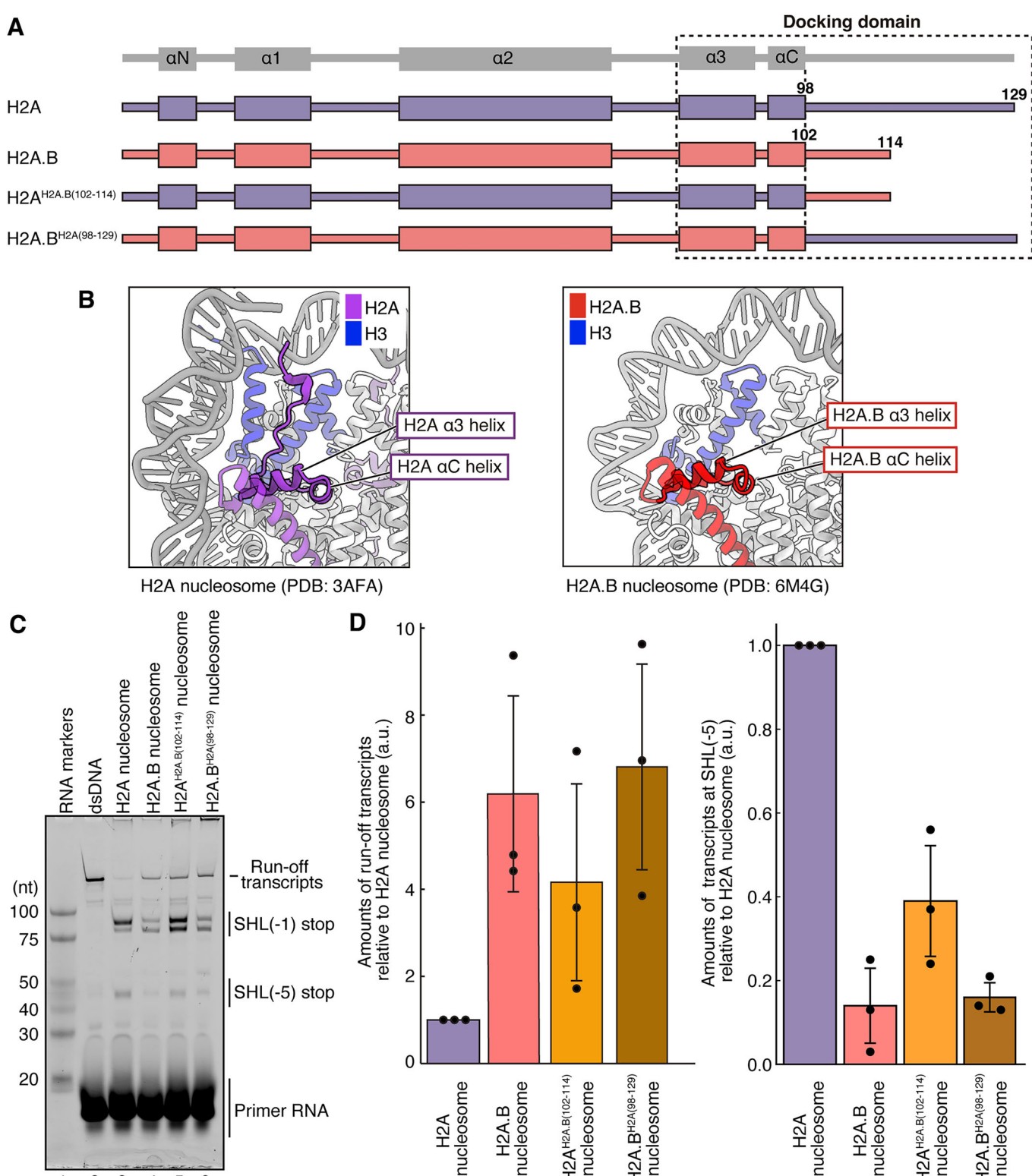

nucleosome might reduce the chance of nucleosome reassembly, leading to nucleosome loss after the EC passage. Further studies are required to solve this issue.

Why does the proximal H2A.B–H2B dimer dissociation occur when the EC transcribes the proximal half of the nucleosomal DNA? This may happen because H2A.B lacks the conserved amino acid sequence that forms the docking domain, by which the C-terminal region of H2A directly binds to the H3 N-terminal region in the nucleosome. We previously reported that the shorter C-terminal region of H2A.B is responsible for the association with

**Figure 4. The H2A.B C-terminal region is responsible for efficient nucleosome transcription.**

(A) Graphical representation of the chimeric histone between H2A and H2A.B used for the nucleosome transcription assay. (B) Close-up views around the C-terminal regions of H2A (left panel) and H2A.B (right panel) in the nucleosome structures. (C) The nucleosome transcription assay. RNA transcripts were detected with DY647 fluorescent dye. Reproducibility was confirmed by three independent experiments (shown in Appendix Fig. S7A,B). (D) Quantification of the transcription assay. Band intensities of the run-off transcripts (left panel) and transcripts paused at the SHL(−5) position (right panel) were quantitated. Amounts of the transcripts in the H2A.B, H2A$^{H2A.B(102-114)}$, and H2A.B$^{H2A(98-129)}$ nucleosomes were plotted relative to those in the H2A nucleosome (a.u.). Error bars indicate standard deviations ($n = 3$ independent replicates). The central values of the error bars for the run-off and SHL(−5) transcripts (i.e., the mean values of the plots) are 6.19 and 0.14 for the H2A.B nucleosome, 4.16 and 0.39 for the H2A$^{H2A.B(102-114)}$ nucleosome, and 6.81 and 0.16 for the H2A.B$^{H2A(98-129)}$ nucleosome, respectively. Source data are available online for this figure.

the H3-H4 tetramer (Hirano et al, 2021). Intriguingly, the H2A mutant, in which the amino acid residues Val109, Ala110, Pro111, Gly112, Glu113, and Asp114, comprising the docking domain, were replaced by the corresponding H2A.B residues, reportedly exhibits defective histone octamer formation (Zhou et al, 2021). The H2A docking domain region also functions in the DNA wrapping around the entry/exit regions of the nucleosome by stabilizing the H3 N-terminal region, which directly binds the DNA at the entry/exit regions. Consistently, our mutational analysis indicated that the EC transcription on the nucleosome containing the H2A mutant, in which the C-terminal region is replaced by the corresponding region of H2A.B, is comparable to the EC transcription on the H2A.B nucleosome (Fig. 4C). In fact, this H2A mutation drastically reduces the EC pausing at the SHL(−5) position and increases the amount of run-off transcripts (Fig. 4C). Therefore, the characteristic C-terminal region of H2A.B plays an essential role in enhancing nucleosome transcription. These H2A.B-specific characteristics in nucleosome transcription may be conserved among the H2A variants with short C-terminal regions, such as H2A.L and H2A.P (Govin et al, 2007; Barral et al, 2017; Molaro et al, 2018; Jiang et al, 2020).

It should be noted that the H2A.B C-terminal region is not solely responsible for the enhancement of EC transcription. The H2A.B mutant containing the canonical H2A C-terminal region also exhibited increased EC transcription efficiency (Fig. 4C; Appendix Fig. S7A,B). Previous studies have shown that, aside from the H2A.B-specific C-terminal residues, several residues in the histone-fold domain, including Gln33, Glu35, Glu40, His42, Gln45,

Glu73, Gly78, Glu79, Leu85, Leu86, Met89, Val90, and His92, contribute to nucleosome stability, as assessed by micrococcal nuclease susceptibility (Zhou et al, 2021). These residues may also play a role in enhancing transcription efficiency in the nucleosome. Further studies are needed to confirm these findings.

H2A.B has been proposed to function in DNA repair and replication, because it accumulates on damaged DNA and replicating DNA regions in the genome (Arimura et al, 2013; Sansoni et al, 2014). In the present study, we report that the H2A.B-H2B dimer is dynamically released during transcription elongation on the nucleosome. The H2A.B nucleosome formed at damaged sites and/or replicating sites of the genome may also enhance the activities of DNA repair and replication enzymes by the dissociating propensity of the H2A.B-H2B dimer. It will be intriguing to study these phenomena in the future by coupling the cryo-EM strategy with in vitro DNA repair and/or replication reactions.

# Methods

## Histone preparation

Human histones H2A, H2B, H3.3, H4, H2A.B, H2A$^{H2A.B(102-114)}$, and H2A.B$^{H2A(98-129)}$ were purified by the methods reported previously (Kujirai et al, 2018b; Arimura et al, 2013; Hirano et al, 2021). Briefly, each histone, H2A, H2B, H3.3, or H2A$^{H2A.B(102-114)}$, was

**Reagents and tools table**

| Reagent/resource | Reference or source | Identifier or catalog number |
|---|---|---|
| **Experimental models** | | |
| *E. coli* (BL21(DE3)) | Merck | Cat#69450 |
| *E. coli* (BL21(DE3)) | Bio-Rad | Cat#69450 |
| *E. coli* (JM109) | Promega | Cat#P9801 |
| *E. coli* (BL21-Codon Plus(DE3)-RIL) | Agilent | Cat#P9801 |
| *E. coli* (DH5α) | Takara | Cat#9057 |
| *E. coli* (NEB Turbo) | New England Biolabs | Cat#C2984I |
| *E. coli* (KRX) | Promega | Cat#L3002 |
| *K. phaffi* (THY46 strain) | Higo et al, 2014 | N/A |
| **Recombinant DNA** | | |
| pET-15b-H2A | Machida et al, 2018 | N/A |
| pET-15b-H2B | Machida et al, 2018 | N/A |
| pET-15b-H3.3 | Machida et al, 2018 | N/A |

| Reagent/resource | Reference or source | Identifier or catalog number |
|---|---|---|
| pET-15b-H4 | Machida et al, 2018 | N/A |
| pHCE-H2A.B | Arimura et al, 2013 | N/A |
| pET-15b-H2A^H2A.B(102-114) | Hirano et al, 2021 | N/A |
| pET-15b-H2A.B^H2A(98-129) | This study | N/A |
| pGEM-T Easy- 153 bp modified Widom 601 sequence | Kujirai et al, 2018a | N/A |
| pET-47b-TFIIS | Ehara et al, 2017b | N/A |
| pET-47b-Spt4/5 | Ehara et al, 2017b | N/A |
| pET-47b-Elf1 | Ehara et al, 2017b | N/A |
| **Antibodies** | | |
| N/A | N/A | N/A |
| **Oligonucleotides and other sequence-based reagents** | | |
| 42 nt DNA fragment containing the 9 base-pair mismatched region | Kujirai et al, 2018a; FASMAC | CCCAAACACACCAAACACAAGAGCTAATTGACTGACGTAAGC |
| 45 nt DNA fragment containing the 9 base-pair mismatched region: | Kujirai et al, 2018a; FASMAC | GCTTACGTCAGTCTGGCCATCTTTGTGTTTGGTGTGTTTGGGTGG |
| 11 nt primer RNA conjugated DY647 florescent dye: | Kujirai et al, 2018a; Dhamacon | DY647-AUAAUUAGCUC |
| 24 nt oligonucleotide for the nucleosomal DNA amplification (non-template strand) | This study; FASMAC | CGAAGGCCGTGGTGGCCGTTTTCG |
| 27 nt oligonucleotide for the nucleosomal DNA amplification conjugated 6-FAM fluorescent dye (template strand) | This study; FASMAC | 6-FAM-GATATCAGAATCCCGGTGCCGAGGCCG |
| 45 nt DNA fragment conjugated Texas-RED fluorescent dye: | Kujirai et al, 2018a; This study; FASMAC | TexasRED- GCTTACGTCAGTCTGGCCATCTTTGTGTTTGGTGTGTTTGGGTGG |
| 11 nt primer RNA: | Kujirai et al, 2018a; This study, FASMAC | AUAAUUAGCUC |
| **Chemicals, enzymes and other reagents** | | |
| Thrombin protease | Wako | Cat# 206-18411 |
| Alkaline Phosphatase (Calf intestine) (CIAP) | Takara | Cat#2250A |
| Glutaraldehyde 25% solution, practical grade | Electron Microscopy Sciences | Cat#16220-P |
| Nonidet® P40 Substitute | Nacalai | Cat#18551-24 |
| SYBR™ Gold Nucleic Acid Gel Stain | Thermo Fisher Scientific | Cat#S11494 |
| T4 DNA Ligase | NIPPON GENE | Cat#317-00406 |
| *Eco*RV restriction enzyme | Takara | Cat#1042A |
| *Bgl*I restriction enzyme | Takara | Cat#1020A |
| Proteinase K, recombinant, PCR Grade | Roche | Cat#03115844001 |
| PrimeSTAR® GXL DNA Polymerase | Takara | Cat#R050A |
| Hi-Di™ Formamide | Applied Biosystems™ | Cat# 4311320 |
| DNase I | Fujifilm Wako | Cat#043-26773 |
| Ban I | New England Biolabs | Cat# R0118L |
| MfeI-HF | New England Biolabs | Cat#R3589S |
| **Software** | | |
| Serial EM ver. 3 | Mastronarde, 2005 | https://bio3d.colorado.edu/SerialEM/ |
| Relion 3.1 | Zivanov et al, 2018 | https://www3.mrc-lmb.cam.ac.uk/relion//index.php/Main_Page |
| Relion 4.0 | Kimanius et al, 2021 | https://www3.mrc-lmb.cam.ac.uk/relion//index.php/Main_Page |

| Reagent/resource | Reference or source | Identifier or catalog number |
|---|---|---|
| Relion 5.0 | Kimanius et al, 2024 | https://www3.mrc-lmb.cam.ac.uk/relion//index.php/Main_Page |
| Motion Corr2 1.4.0 | Zheng et al, 2017 | https://emcore.ucsf.edu/ucsf-software |
| CTFFind 4.1.14 | Rohou and Grigorieff, 2015 | https://grigorieflab.umassmed.edu/ctf_estimation_ctffind_ctftilt |
| UCSF Chimera 1.17.3 | Pettersen et al, 2004 | https://www.cgl.ucsf.edu/chimera/ |
| UCSF Chimera X 1.7.1 | Goddard et al, 2018 | https://www.cgl.ucsf.edu/chimerax/ |
| PyMOL™ 2.4.0 | Schrodinger, LLC | https://www.pymol.org/ |
| Coot 0.8.9 | Emsley et al, 2010 | https://www2.mrc-lmb.cam.ac.uk/personal/pemsley/coot/ |
| Phenix 1.20.1 | Liebschner et al, 2019 | https://phenix-online.org/ |
| ISOLDE 1.3 | Croll, 2018 | https://tristanic.github.io/isolde/ |
| Mol Probity | Williams et al, 2018 | http://molprobity.biochem.duke.edu/ |
| ImageJ 1.52a | Schneider et al, 2012 | https://imagej.net/ij/index.html |
| ImageQuant™ TL version 8.1.0.0 | Cytiva | N/A |
| **Other** | | |
| Ni-NTA agarose beads | Qiagen | Cat#30250 |
| Mono S HR 16/10 column | Cytiva | Cat#17-0507-01 |
| Q Sepharose Fast Flow | Cytiva | Cat#17051001 |
| ANTI-FLAG M2 Magnetic Bead | Sigma | Cat#M8823 |
| Ni Sepharose 6 Fast Flow | Cytiva | Cat#17531802 |
| Resource Q | Cytiva | Cat#17117701 |
| Resource S | Cytiva | Cat#17117801 |
| TSKgel DEAE-5PW | TOSOH | Cat#0007574 |
| HiLoad 16/600 Superdex 200 pg column | Cytiva | Cat#28-9893-35 |
| Amicon Ultra-15 centrifugal filter unit (30,000 Da) | Merck Millipore | Cat#UFC903096 |
| Amicon Ultra-2 centrifugal filter unit (100,000 Da) | Merck Millipore | Cat#UFC210024 |
| Model 491 Prep Cell | Bio-Rad | Cat#1702928 |
| Perista BioMini Pump | ATTO | Cat#1221200 |
| Econo column® chromatography columns | Bio-Rad | Cat#7372512 |
| PD-10 desalting columns | Cytiva | Cat#17085101 |
| NanoDrop™ One | Thermo Fisher Scientific | Cat#ND-ONEC-W |
| Amersham™ Imager 680QC | Cytiva | Cat#29270771 |
| Amersham™ Typhoon™ Scanner | Cytiva | N/A |
| Gradient maker | Biocomp Instruments | N/A |
| SW 41 Ti rotor | Beckman Coulter | Cat#331362 |
| Vitrobot Mark IV | Thermo Fisher Scientific | N/A |
| Quantifoil R1.2/1.3 200-mesh Cu | Quantifoil | Cat#M2955C-1 |
| Krios G4 | Thermo Fisher Scientific | N/A |

produced as the N-terminally His$_6$-tagged protein in the *Eschericha coli* BL21(DE3) strain. H2A.B and H2A.B$^{H2A(98-129)}$ were produced in the BL21(DE3) cpRIL strain. H4 was produced in the JM109 strain. The *E. coli* cells producing each histone were cultivated and the cells were disrupted by sonication. Histones were purified by Ni-NTA affinity chromatography (Qiagen) under denaturing conditions.

For H2A, H2B, H3.3, and H4, the His$_6$-tag portion was removed by thrombin protease cleavage and purified by Mono S cation exchange column chromatography (Cytiva) under denaturing conditions. Finally, the histones were freeze-dried and stored as powders at 4 °C. For reconstitution of the H2A-H2B and H3.3-H4 complexes, the H2A and H2B powders and H3.3 and H4 powders were mixed under denaturing conditions at a molar

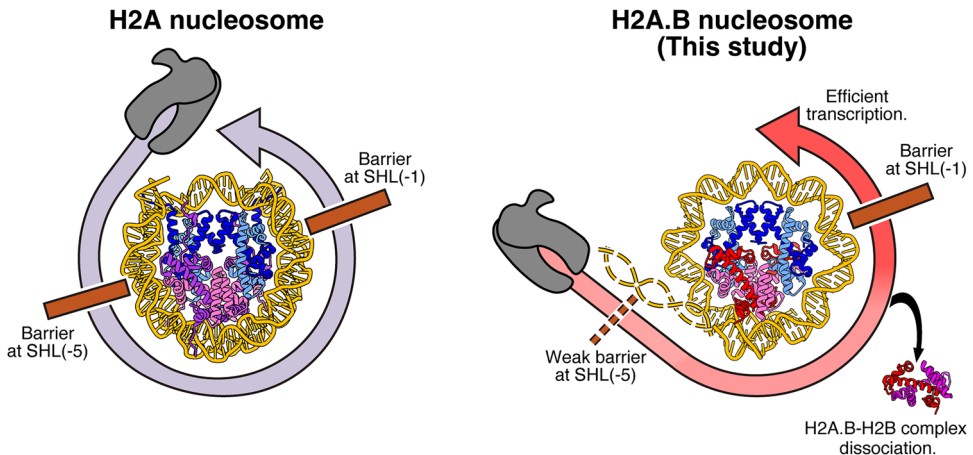

**H2A nucleosome**

**H2A.B nucleosome (This study)**

Efficient transcription.

Barrier at SHL(−1)

Barrier at SHL(−5)

Barrier at SHL(−1)

Weak barrier at SHL(−5)

H2A.B-H2B complex dissociation.

**Figure 5. A model of H2A.B nucleosome transcription.**

During transcription on the canonical H2A nucleosome, RNAPII substantially pauses at the SHL(−5) and SHL(−1) positions (left model). During transcription on the H2A.B nucleosome, the RNAPII efficiently passes through the SHL(−5) position, concomitant with the dissociation of the proximal H2A.B-H2B dimer (right model).

ratio of 1:1. The samples were then dialyzed against high salt buffer, and the resulting complexes were purified by HiLoad 16/60 200 pg gel exclusion column chromatography (Cytiva). The purified samples were frozen at −80 °C. For reconstitution of the H2A.B-H2B, H2A$^{H2A.B(102-114)}$-H2B and H2A.B$^{H2A(98-129)}$-H2B complexes, purified H2A.B, H2A$^{H2A.B(102-114)}$ or H2A.B$^{H2A(98-129)}$ was mixed with H2B under denaturing conditions. The H2A.B-H2B, H2A$^{H2A.B(102-114)}$-H2B and H2A.B$^{H2A(98-129)}$-H2B complexes were then refolded by dialysis against high salt buffer. The His$_6$-tag portion was then removed by thrombin protease cleavage, and the resulting complexes were further purified by HiLoad 16/60 200 pg gel exclusion column chromatography (Cytiva). Purified histone complexes were stored at −80 °C.

## Preparation of the modified Widom 601 DNA nucleosome template

The 153 base-pair DNA containing the modified Widom 601 sequence was purified according to the method previously reported (Kujirai et al, 2018a; Lowary and Widom 1998). Briefly, a 153 base-pair DNA fragment containing the modified Widom 601 sequence was inserted into the pGEM-TE vector. The plasmid was amplified in the *E. coli* DH5α strain. The modified Widom 601 fragment was excised from the plasmid by *Eco*RV cleavage (Takara) and purified by polyethylene glycol precipitation. The purified fragment was dephosphorylated by alkaline phosphatase (Takara) and cleaved with *Bgl*I to create the bubble DNA ligation site (see below). The DNA fragments were purified by DEAE anion exchange column chromatography (TOSOH). The DNA sequences of the final products are:

Non-template strand:

5′;-TGGCCGTTTTCGTTGTTTTTTTCTGTCTCGTGCCTGG
TGTCTTGGGTGTAATCCCCTTGGCGGTTAAAACGCGGGGG
ACAGCGCGTACGTGCGTTTAAGCGGTGCTAGAGCTGTCT
ACGACCAATTGAGCGGCCTCGGCACCGGGATTCTGAT -3′.

Template strand:

5′-ATCAGAATCCCGGTGCCGAGGCCGCTCAATTGGTCG
TAGACAGCTCTAGCACCGCTTAAACGCACGTACGCGCTGT
CCCCCGCGTTTTAACCGCCAAGGGGATTACACCCAAGACA
CCAGGCACGAGACAGAAAAAAACAACGAAAACGGCCA
CCA -3′.

## Nucleosome reconstitution and purification

Nucleosomes were reconstituted and purified by the methods described previously (Kujirai et al, 2018b; Arimura et al, 2013). The purified histone complexes and 153 base-pair DNA were mixed and reconstituted by the salt dialysis method. A 45 base-pair bubble DNA fragment containing a 9 base-pair mismatch region was ligated to the *Bgl*I-cleaved end of the nucleosome DNA by T4 ligase (NIPPON GENE). The resulting nucleosome containing the bubble DNA region was purified by non-denaturing polyacrylamide gel electrophoresis, using a Prep Cell apparatus (Bio-Rad). The purified nucleosome was concentrated and stored at −80 °C.

## Preparations of RNAPII, TFIIS, Spt4/5, and Elf1

*Komagataella phaffii* (formerly *pastoris*) RNAPII was purified by the method described previously, with slight modifications (Higo et al, 2014; Ehara et al, 2017a; Ehara et al, 2017b). In brief, the TAP-tagged Rpb2 subunit of RNAPII was expressed in *Komagataella phaffii* cells. The RNAPII containing the TAP-tagged Rpb2 was purified by chromatography on Q Sepharose Fast Flow anion exchange (Cytiva), anti-FLAG M2 affinity gel, and Resource Q anion exchange (Cytiva) columns.

The transcription elongation factors, TFIIS, Spt4, Spt5, and Elf1, were produced as His$_6$-tagged recombinant proteins in *E. coli* cells and purified by COSMOGEL His-Accept column (Nacalai Tesque) affinity chromatography (Ehara et al, 2017b). The His$_6$-tag portion was removed by HRV-3C protease cleavage, and the resulting protein without the His$_6$-tag was purified by Resource S (for TFIIS) or Resource Q (for Spt4/5 and Elf1) ion exchange column chromatography.

## Transcription assay

In vitro transcription experiments were performed by the previously described methods (Kujirai et al, 2018a; Ehara et al, 2019). For in vitro transcription reactions, 0.1 μM RNAPII was mixed with 0.1 μM nucleosome, 0.1 μM TFIIS, 0.4 μM Spt4/5, 1.0 μM Elf1, and 0.4 μM primer RNA conjugated with DY647 fluorescent dye in 28 mM HEPES-KOH buffer (pH 7.5), containing 50 mM $CH_3COOK$, 5 mM $MgCl_2$, 0.1 μM $(CH_3COO)_2Zn$, 30 μM TCEP-HCl, 0.4 mM UTP, 0.4 mM CTP, 0.4 mM GTP, 0.4 mM ATP, 2% glycerol, and 0.1 mM DTT. The transcription reaction was conducted at 4 °C for 4 min. The samples were then deproteinized with proteinase K (Roche), incubated with HiDi Formamide (Thermo Fisher Scientific) at 95 °C to denature the RNA transcripts, and subjected to denaturing 10% polyacrylamide gel electrophoresis. The DY647 signal on the gel was detected using a Typhoon imager (Cytiva). To ensure the consistency of the results, each transcription assay was independently repeated three times. The band intensities of RNA transcripts were quantified and normalized using the ImageQuant™ TL software, version 8.1.0.0 (Cytiva). The ratios of the band intensities were compared with those of the canonical H2A nucleosome.

## Preparation of the Spt4/5-Elf1-RNAPII-H2A.B nucleosome complexes for cryo-EM analysis

The Spt4/5-Elf1-RNAPII-H2A.B nucleosome complexes for cryo-EM analysis were prepared as described previously (Kujirai et al, 2018a; Ehara et al, 2019). The nucleosome transcription reaction was conducted with 0.1 μM RNAPII, 0.1 μM nucleosomes, 0.1 μM TFIIS, 0.4 μM Spt4/5, 1.0 μM Elf1, and 0.4 μM primer RNA conjugated with DY647 fluorescent dye in 28 mM HEPES-KOH buffer (pH 7.5), containing 50 mM $CH_3COOK$, 5 mM $MgCl_2$, 0.1 μM $(CH_3COO)_2Zn$, 30 μM TCEP-HCl, 2% glycerol, and 0.1 mM DTT. For the SHL(−5) paused sample, the transcription reaction was conducted in the presence of 0.4 mM UTP and 0.4 mM GTP. For the SHL(−1) paused sample, the transcription reaction was conducted in the presence of 0.4 mM UTP, 0.4 mM GTP, and 0.4 mM CTP. The reaction mixtures were incubated for 30 min at 30 °C, and then quenched with 50 mM EDTA. These samples were fractionated by sucrose density gradient ultracentrifugation coupled with a glutaraldehyde gradient (GraFix method) (Kastner et al, 2008). The sucrose gradient solution was prepared with low sucrose solution (20 mM HEPES-KOH (pH 7.5), 0.2 μM $(CH_3COO)_2Zn$, 0.1 mM TCEP-HCl, and 10% sucrose) and high sucrose solution (20 mM HEPES-KOH (pH 7.5), 0.2 μM $(CH_3COO)_2Zn$, 0.1 mM TCEP-HCl, 25% sucrose, and 0.1% glutaraldehyde), using a Gradient Master instrument (Biocomp Instruments). The reaction mixture was loaded on the top of the sucrose gradient solution and ultracentrifuged at 27,000 rpm at 4 °C for 16 h, using an SW41 rotor (Beckman Coulter). The fractions containing the Spt4/5-Elf1-RNAPII- nucleosome complexes were desalted with a PD-10 column (Cytiva), pre-equilibrated with 20 mM HEPES-KOH (pH 7.5), 0.2 μM $(CH_3COO)_2Zn$, and 0.1 mM TCEP-HCl, and concentrated using an Amicon Ultra 100 K filter (Millipore).

For the cryo-EM analysis, before the sample plunging, 0.003% NP-40 was added to improve the particle orientation distribution. The final concentrations of the Spt4/5-Elf1-RNAPII-H2A.B nucleosome paused at the SHL(−5) position and the SHL(−1) position were 80 ng/μl and 60 ng/μl, respectively. The samples (2.0 μL) were applied to glow-discharged Quantifoil R1.2/1.3 200-mesh grids (Quantifoil) and plunged using a Vitrobot Mark IV (Thermo Fisher Scientific) at 4 °C and 100% humidity.

## Cryo-EM data collection and image processing

Cryo-EM images for the Spt4/5-Elf1-RNAPII-H2A.B nucleosome complexes were acquired using a Krios G4 cryo-electron microscope (Thermo Fisher Scientific) equipped with a K3 direct electron detector and a BioQuantum energy filter (Gatan) with a slit width of 20 eV. Details of the data acquisition are shown in Table 1. In total, 9786 movies for the Spt4/5-Elf1-RNAPII-H2A.B nucleosome paused at the SHL(−5) position were imaged using the EPU software (Thermo Fisher Scientific) and a total of 23,935 movies for the Spt4/5-Elf1-RNAPII-H2A.B nucleosome paused at the SHL(−1) position were imaged using SerialEM software (Mastronarde, 2005). For data acquisition, the pixel size was 1.06 Å, and the defocus values were between −1.0 and −2.5 μm. The dose per frame was 1.431 e/Å$^2$ for the Spt4/5-Elf1-RNAPII-H2A.B nucleosome paused at the SHL(−5) position. The doses per frame were 1.451 e/Å$^2$ for the Spt4/5-Elf1-RNAPII-H2A.B nucleosome paused at the SHL(−1) position (dataset 1) and 1.439 e/Å$^2$ (dataset 2). The motion correction and CTF estimation were conducted using MotionCor2 (Zheng et al, 2017) and CTFFIND4 (Rohou and Grigorieff, 2015), respectively. All image processing was executed by RELION 3.1 (Zivanov et al, 2018), RELION 4.0 (Kimanius et al, 2021), and Relion 5.0 (Kimanius et al, 2024). The image processing workflow is shown in Figs. EV1 and EV3.

For the image processing of the Spt4/5-Elf1-RNAPII-H2A.B nucleosome paused at the SHL(−5) position, the reference particles for Topaz picking (Bepler et al, 2019) were picked by auto-picking based on a Laplacian-of-Gaussian (LoG) filter. These particles were 2D/3D classified, and a reference class average (6.6k particles) was selected. Topaz picking was performed, and about 3 million particles were re-picked. These particles were extracted with a binning factor of 4 (pixel size of 4.24 Å/pixel), and bad particles were removed through 2D and 3D classifications.

Overall 3D classification was performed twice to select the classes containing RNAPII and nucleosome densities, and the selected classes were re-extracted (pixel size of 1.06 Å/pixel). The EM densities of Spt4/5-Elf1-RNAPII were subtracted from particles of the Spt4/5-Elf1-RNAPII-H2A.B nucleosome complex, followed by further 3D classification of the nucleosome. After the classification, seven classes (110,782 particles) containing better nucleosome densities were selected, and 3D refinement and postprocessing were performed. These particles were reverted by subtraction, and 3D refinement and postprocessing were performed. The final resolutions of the post-processed maps of the Spt4/5-Elf1-RNAPII-H2A.B nucleosome paused at the SHL(−5) position were 4.3 Å (overall), 4.2 Å (RNAPII region), and 6.9 Å (nucleosome region) (FSC = 0.143) (Scheres, 2016).

Particles of the Spt4/5-Elf1-RNAPII-H2A.B nucleosome paused at the SHL(−1) position were picked by auto-picking based on a Laplacian-of-Gaussian (LoG) filter for each dataset. Particles were extracted with a binning factor of 2 (pixel size of 2.12 Å/pixel) and sorted by 2D classification. After the 2D classifications, the good average classes from each dataset were merged, and the 3D classes

**Table 1.  Cryo-EM data collection and image processing.**

| Sample | Spt4/5-Elf1-RNAPII-H2A.B nucleosome (SHL(−5)stop) | Spt4/5-Elf1-RNAPII-H2A.B nucleosome (SHL(−1)stop) |
|---|---|---|
| PDB ID | N/A | 9II7 |
| EMDB ID | EMD-60592 | EMD-60593 |
| **Data collection** | | |
| Electron microscope | Krios G4 (Thermo Fisher Scientific) | |
| Camera | K3 BioQuantum (Gatan) | |
| Pixel size (Å/pix) | 1.06 | |
| Defocus range (μm) | −1.0 to −2.5 | |
| Number of frames | 40 | |
| Dose per frame (e⁻/Å²/frame) | 1.431 | 1.451 (dataset 1) 1.439 (dataset 2) |
| Number of collected micrographs | 9786 | 10,030 (dataset 1) 13,905 (dataset 2) |
| Number of selected micrographs | 8852 | 9,444 (dataset 1) 12,693 (dataset 2) |
| **Reconstruction** | | |
| Number of picked particles | 3,029,728 | 2,023,692 (dataset 1) 2,845,742 (dataset 2) |
| Number of particles used for refinement | 110,782 | 297,456 |
| Symmetry applied | C1 | C1 |
| Final resolution (overall) (Å) | 4.3 | 3.5 |
| Final resolution (RNAPII) (Å) | 4.2 | 3.2 |
| Final resolution (nucleosome) (Å) | 6.9 | 4.8 |
| FSC threshold | 0.143 | 0.143 |
| **Validation** | | |
| MolProbity score | N/A | 1.90 |
| Clash score | N/A | 13.28 |
| RMSDs | | |
| Bond lengths (Å) | N/A | 0.008 |
| Bond angles (°) | N/A | 1.258 |
| Ramachandran plot (%) | | |
| Outliers | N/A | 0.13 |
| Allowed | N/A | 3.84 |
| Favored | N/A | 94.04 |
| Rotamer outliers (%) | N/A | 0.70 |

were further classified to remove bad particles. Finally, 297,456 particles containing Spt4/5, Elf1, RNAPII, and H2A.B nucleosome densities were selected, rescaled (pixel size of 1.06 Å/pixel), and 3D refined. Each Spt4/5-Elf1-RNAPII or H2A.B nucleosome region was extracted from this density map, and further focused refinements were performed and postprocessed. The final resolutions of the post-processed maps of the Spt4/5-Elf1-RNAPII-H2A.B nucleosome paused at the SHL(−1) position were 3.5 Å

(overall), 3.2 Å (RNAPII region), and 4.8 Å (nucleosome region) (FSC = 0.143) (Scheres, 2016).

The composite maps were created from local subtracted Spt4/5-Elf1-RNAPII and nucleosome maps, using phenix.combine_focused_maps in the Phenix software version 1.20.1 (Liebschner et al, 2019).

## Model building

The model for the Spt4/5-Elf1-RNAPII-H2A.B nucleosome at the SHL(−1) position was built based on the cryo-EM densities of the overall map, and the focused RNAPII and focused H2A.B nucleosome maps. The Spt4/5-Elf1-RNAPII-DNA region was initially built and refined using the Spt4/5-Elf1-RNAPII-H2A nucleosome (SHL(−1) stop) complex (PDB ID: 6IR9) (Ehara et al, 2019), and nucleosomal histone regions were based on the H2A.B nucleosome structure (PDB ID: 6M4G) (Zhou et al, 2021). Models were combined using UCSF Chimera (Pettersen et al, 2004) and PyMOL (Schrödinger) and further fixed using Coot (Emsley et al, 2010) and Isolde (Croll, 2018). All cryo-EM map and protein atomic model figures were prepared using UCSF Chimera (Pettersen et al, 2004) and ChimeraX (Goddard et al, 2018).

## Fluorescently labeled nucleosome preparation and DNase I footprinting analysis

The fluorescently labeled nucleosome was prepared as described previously (Arimura et al, 2013, Kujirai et al, 2018b, Ehara et al, 2022). The 153 base-pair DNA fragment was amplified by PCR, using the 6-FAM-labeled primer (FASMAC) (Ehara et al, 2022). The nucleosome was reconstituted by the salt-dialysis method, as described above. After the nucleosome reconstitution, a 45 base-pair Texas-Red labeled bubble DNA fragment (FASMAC) was ligated to one end of the nucleosomal DNA, using T4 DNA ligase (NIPPON GENE). The nucleosome containing the bubble DNA fragment was purified by non-denaturing polyacrylamide gel electrophoresis, using a Prep Cell apparatus (Bio-Rad). The final products were concentrated and stored at −80 °C.

For the transcription reaction, the nucleosome with 6-FAM (0.1 μM) was mixed with 0.1 μM RNAPII, 0.1 μM TFIIS, 0.4 μM Spt4/5, 1.0 μM Elf1, and 0.2 μM primer RNA in 28 mM HEPES-KOH buffer (pH 7.5), containing 50 mM CH₃COOK, 5 mM MgCl₂, 0.1 μM (CH₃COO)₂Zn, 30 μM TCEP-HCl, 0.4 mM UTP, 0.4 mM CTP, 0.4 mM GTP, 50 μM 3′-dATP, 2% glycerol, and 0.1 mM DTT (Ehara et al, 2019), and incubated at 30 °C for 30 min. NTP(-) samples were also incubated under the same conditions, except for the nucleotide composition. After the incubation, 0.04 units (for dsDNA) or 0.05 units (for nucleosomes) of DNase I (Fujifilm Wako) were added to the transcription reaction mixture and further incubated at 30 °C for 10 min. The samples were deproteinized by proteinase K (Roche), incubated with HiDi Formamide (Thermo Fisher Scientific) at 95 °C, and subjected to denaturing 8% polyacrylamide gel electrophoresis. The DNA markers were created from the 198 base-pair 6-FAM-labeled template DNA by restriction enzyme digestions with *Ban*I, *Bgl*I and *Mfe*I. The 6-FAM fluorescence signals were detected using a Typhoon imager (Cytiva). The DNase I footprinting was independently repeated two times.

## The use of large language models

ChatGPT, Grammarly, and DeepL were used for the grammatical correction of the text. No original sentences were produced by AI.

## Data availability

The cryo-EM maps and atomic models have been deposited in the Electron Microscopy Data Bank (EMDB) and the Protein Data Bank (PDB), respectively. The accession codes are EMD-60592 (Spt4/5-Elf1-RNAPII-H2A.B nucleosome(SHL(−5)stop)) and EMD-60593 and 9II7 (Spt4/5-Elf1-RNAPII-H2A.B nucleosome(SHL(−1)stop)).

The source data of this paper are collected in the following database record: biostudies:S-SCDT-10_1038-S44318-025-00473-6.

## Peer review information

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

    determination in RELION-3. eLife 7:e42166

## Acknowledgements

We thank Y Ikura, M Dacher, and Y Takeda (The University of Tokyo) for their
assistance, and Dr. M Kikkawa for cryo-EM data collection at the cryo-EM
facility, The University of Tokyo. We thank M Goto and M Henmi (RIKEN) for
their assistance with protein purification. This work was supported in part by
JSPS KAKENHI Grant Numbers JP22K15033 [to TK], JP22K06098 [to YT],
JP23K20300 [to HE], JP23K17392 [to TK], JP24H00062 [to SS and
TK], JP23H05475, JP24H02319 and JP24H02328 [to HK]; JST SPRING Grant
Number JPMJSP2108 [to MA], JST ERATO Grant Number JPMJER1901 [to
HK]; JST CREST Grant Number JPMJCR24T3 [to HK]; and the Platform Project
for Supporting Drug Discovery and Life Science Research (BINDS) from AMED
under Grant Number JP25ama121009 [to HK] and JP25ama121002 [to M
Kikkawa].

## Author contributions

**Munetaka Akatsu**: Resources; Formal analysis; Funding acquisition;
Investigation; Visualization; Methodology; Writing—review and editing.
**Rina Hirano**: Resources; Investigation; Methodology; Writing—review and
editing. **Tomoya Kujirai**: Resources; Formal analysis; Supervision; Funding
acquisition; Investigation; Methodology; Writing—review and editing. **Mitsuo
Ogasawara**: Data curation; Investigation. **Haruhiko Ehara**: Resources; Formal
analysis; Funding acquisition; Investigation; Visualization; Methodology;
Writing—review and editing. **Shun-ichi Sekine**: Resources; Supervision; Funding
acquisition; Project administration; Writing—review and editing. **Yoshimasa
Takizawa**: Formal analysis; Supervision; Funding acquisition; Writing—review
and editing. **Hitoshi Kurumizaka**: Conceptualization; Supervision; Funding
acquisition; Visualization; Writing—original draft; Project administration;
Writing—review and editing.

Source data underlying figure panels in this paper may have individual
authorship assigned. Where available, figure panel/source data authorship is
listed in the following database record: biostudies:S-SCDT-10_1038-S44318-
025-00473-6.

## Disclosure and competing interests statement

The authors declare no competing interests.

# Expanded View Figures

**Figure EV1. Cryo-EM analysis of the EC-H2A.B nucleosome complex paused at the SHL(−5) position.**

(A) Representative micrograph of the EC-H2A.B nucleosome complex paused at the SHL(−5) position. Scale bar: 100 nm. (B) Representative 2D class averages for the final map of the EC-H2A.B nucleosome complex paused at the SHL(−5) position. Scale bar: 10 nm. (C) Workflow of the image processing of the EC-H2A.B nucleosome complex paused at the SHL(−5) position.

▶

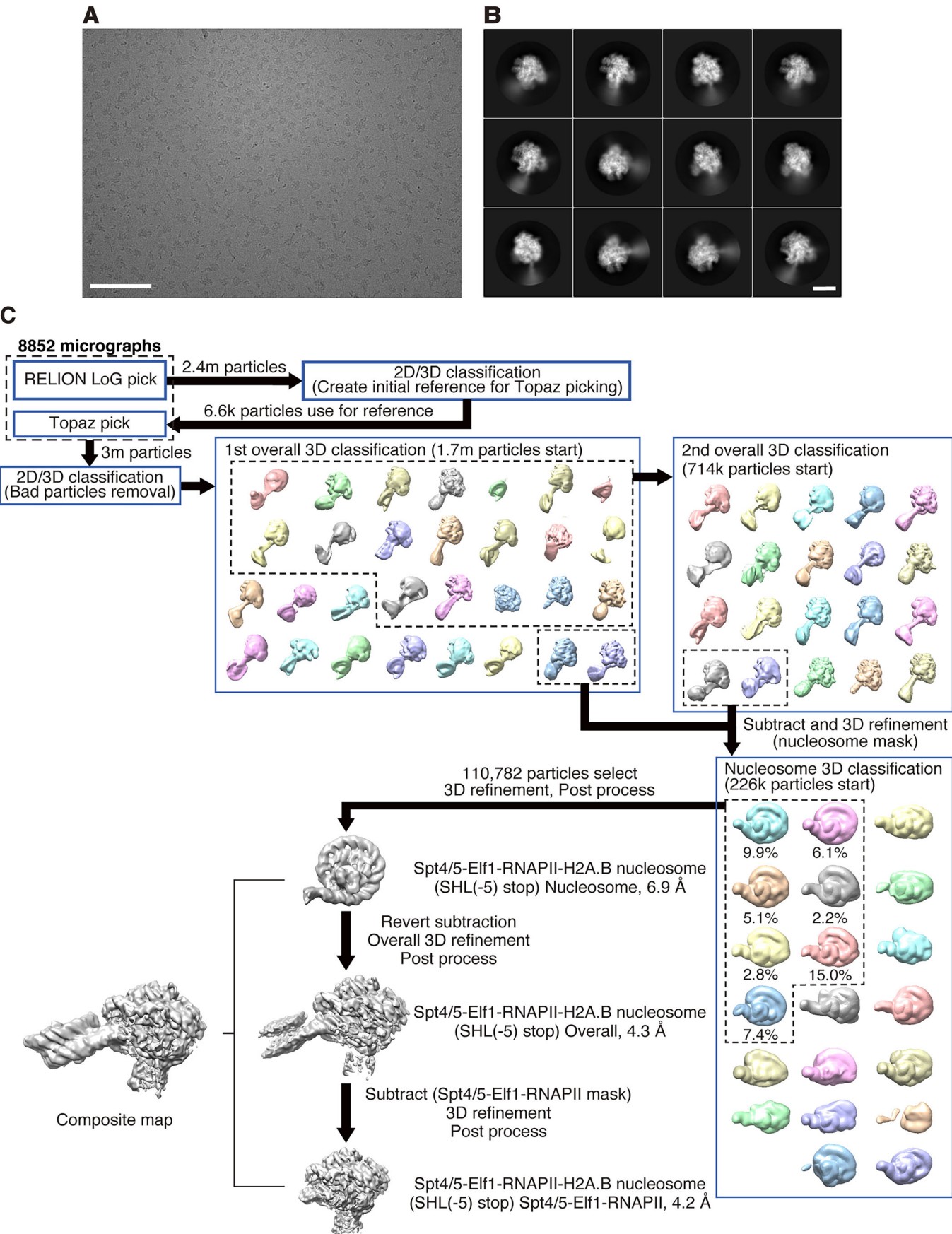

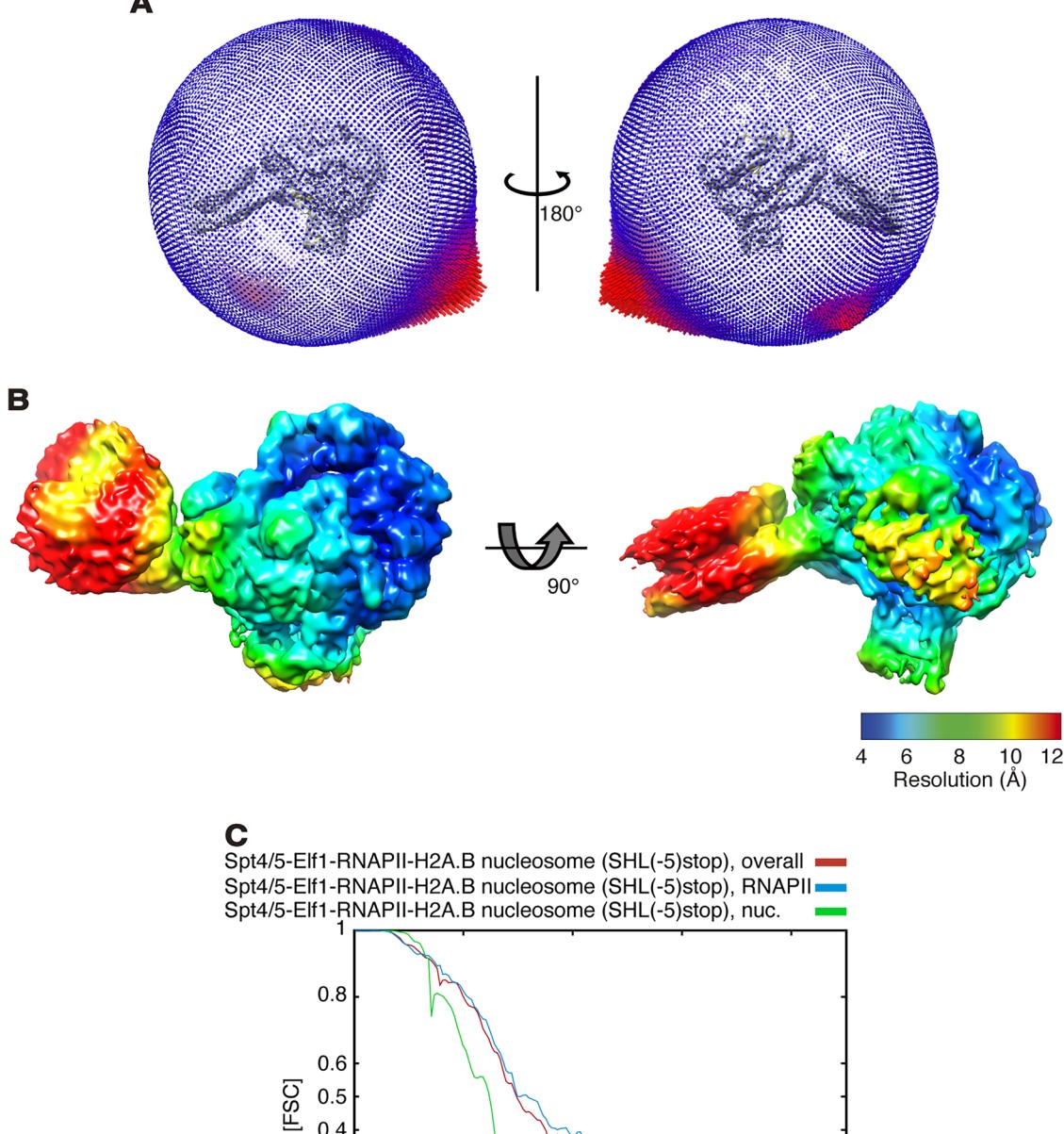

**Figure EV2.  Cryo-EM map and model qualities of the EC-H2A.B nucleosome complex paused at the SHL(−5) position.**

(A) Angular distribution of the EC-H2A.B nucleosome complex paused at the SHL(−5) position. (B) Local resolution map of the EC-H2A.B nucleosome complex paused at the SHL(−5) position. (C) Fourier Shell Correlation (FSC) curve of the EC-H2A.B nucleosome complex paused at the SHL(−5) position. The final resolutions of the structures were estimated at 4.3 Å (overall), 4.2 Å (Spt4/5-Elf1-RNAPII), and 6.9 Å (nucleosome) (FSC = 0.143).

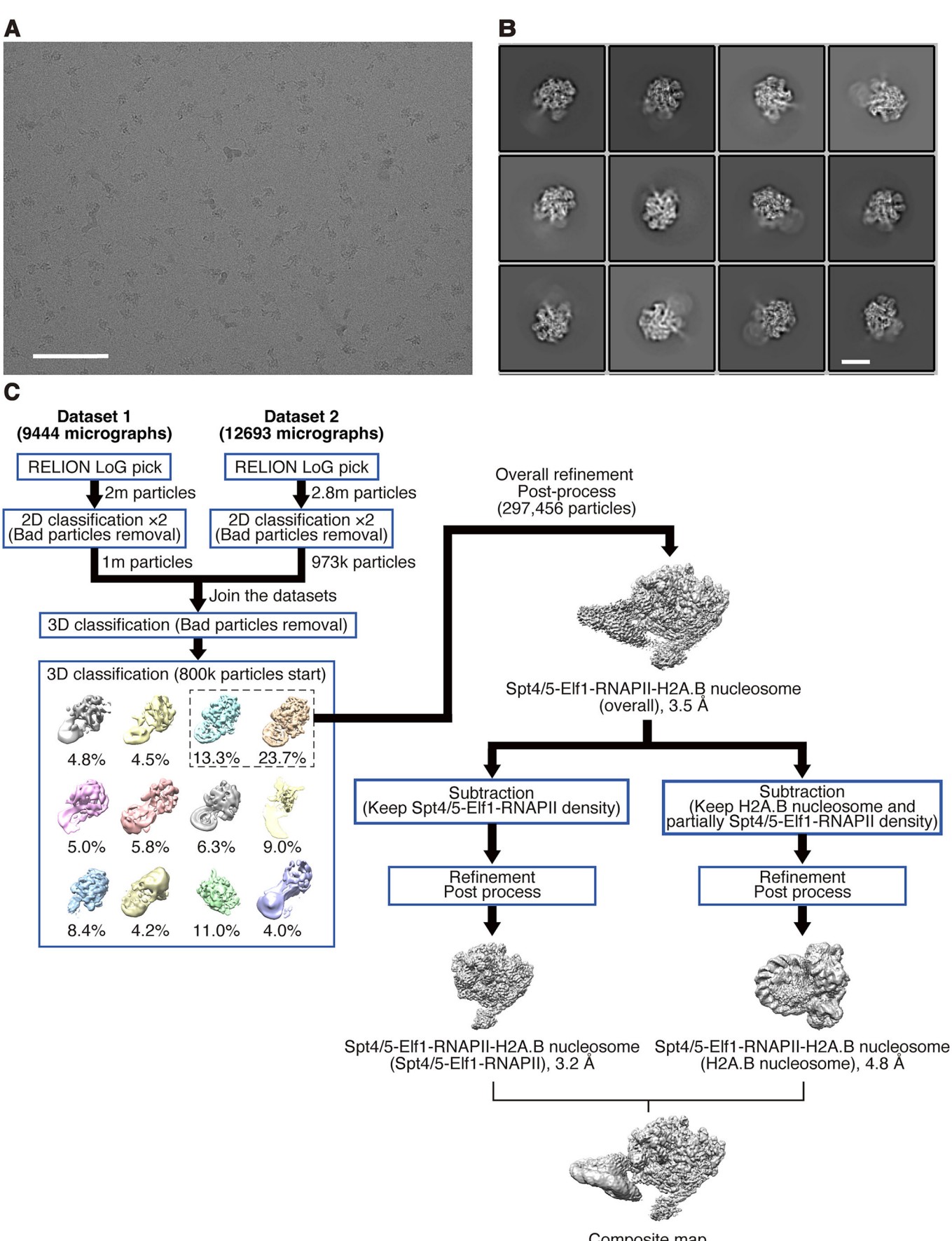

**Figure EV3.  Cryo-EM analysis of the EC-H2A.B nucleosome complex paused at the SHL(−1) position.**

(**A**) Representative micrograph of the EC-H2A.B nucleosome complex paused at the SHL(−1) position. Scale bar: 100 nm. (**B**) Representative 2D class averages for the final map of the EC-H2A.B nucleosome complex paused at the SHL(−1) position. Scale bar: 10 nm. (**C**) Workflow of the image processing of the EC-H2A.B nucleosome complex paused at the SHL(−1) position.

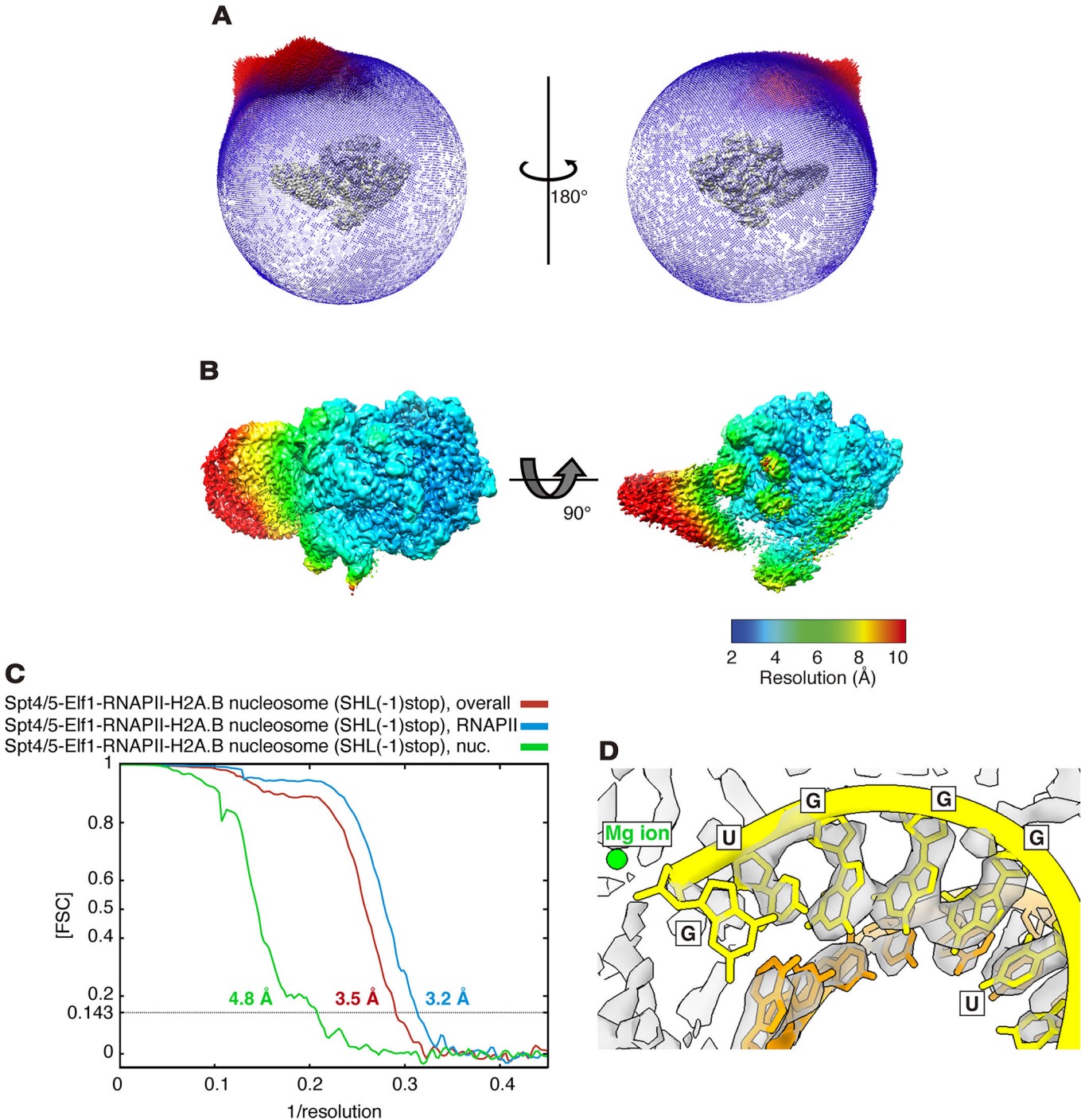

**Figure EV4.  Cryo-EM map and model qualities of the EC-H2A.B nucleosome complex paused at the SHL(−1) position.**

(A) Angular distribution plots of the EC-H2A.B nucleosome complex paused at the SHL(−1) position. (B) Local resolution map of the EC-H2A.B nucleosome complex paused at the SHL(−1) position. (C) Fourier Shell Correlation (FSC) curve of the EC-H2A.B nucleosome complex paused at the SHL(−1) position. The final resolutions of the structures were estimated at 3.5 Å (overall), 3.2 Å (Spt4/5-Elf1-RNAPII), and 4.8 Å (nucleosome) (FSC = 0.143). (D) Close-up view of the RNA in the RNAPII catalytic center. Nascent RNA, template DNA, and magnesium atom are colored yellow, orange, and green, respectively. The RNAPII pausing site is determined based on the RNA sequence (5′-UGGGUG-3′) shown in the panel.

SHL(-5) stop, Contour level: 0.037

Distal H2A.B-H2B dimer

Proximal H2A.B-H2B dimer

SHL(-5) stop, Contour level: 0.031

SHL(-1) stop, Contour level: 0.063

Distal H2A.B-H2B dimer

Proximal H2A.B-H2B dimer

SHL(-1) stop, Contour level: 0.057

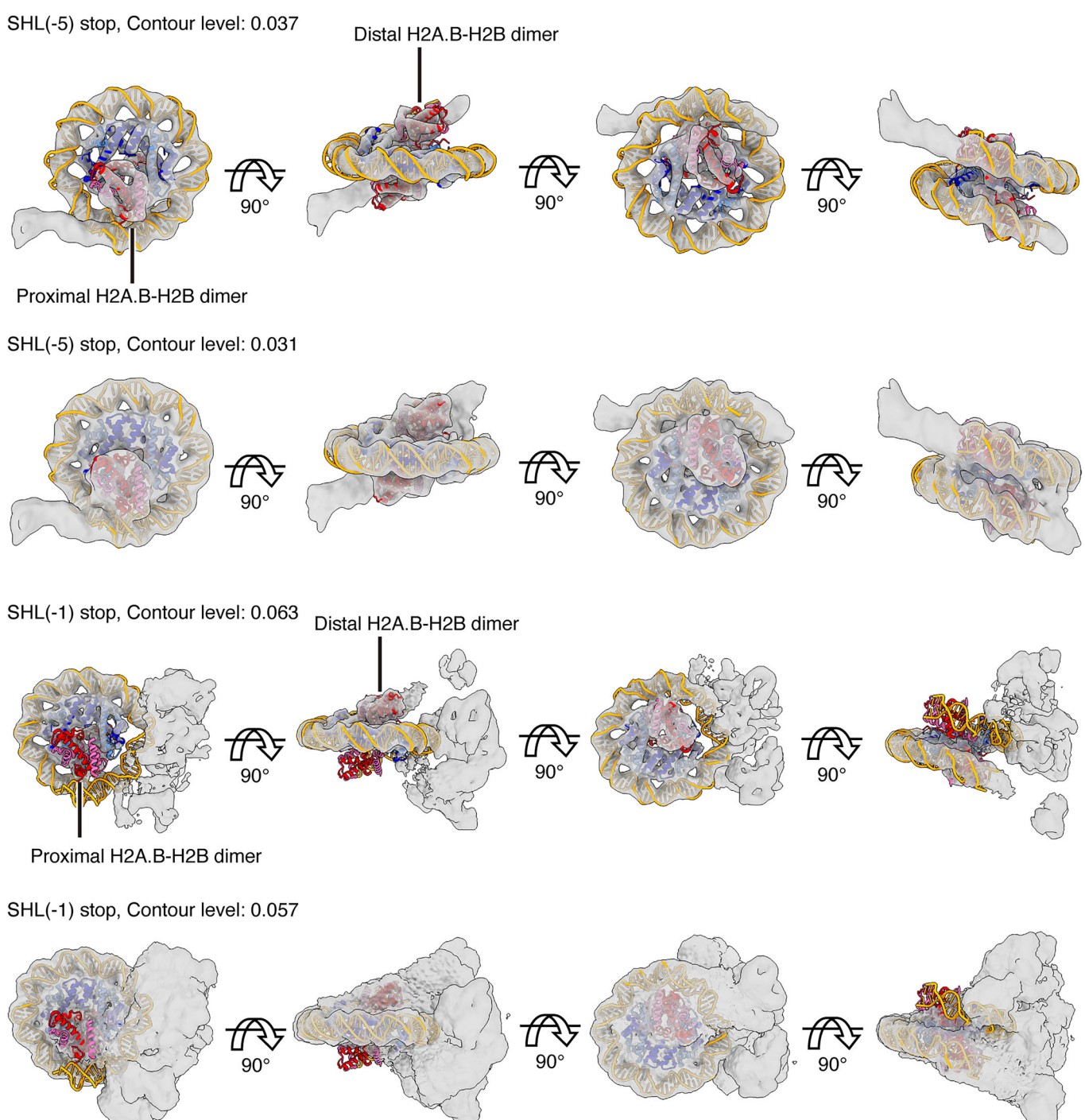

**Figure EV5. Cryo-EM maps of the nucleosome regions in the EC-H2A.B nucleosome complexes paused at the SHL(−5) and SHL(−1) positions.**

Focused refinement maps of the nucleosome regions with different contour levels. The atomic model of the H2A.B nucleosome (PDB: 6M4G) is superimposed.

