## [Peer Review File · The EMBO Journal]

Structural basis of RNAPII transcription on the nucleosome containing histone variant H2A.B

Munetaka Akatsu, Rina Hirano, Tomoya Kujirai, Mitsuo Ogasawara, Haruhiko Ehara, Shun-ichi Sekine, Yoshimasa Takizawa, and Hitoshi Kurumizaka

Corresponding author(s): Hitoshi Kurumizaka (kurumizaka@iqb.u-tokyo.ac.jp)

Review Timeline:

Submission Date:	28th Sep 24
Editorial Decision:	4th Nov 24
Revision Received:	18th Mar 25
Editorial Decision:	10th Apr 25
Revision Received:	15th Apr 25
Accepted:	29th Apr 25

Editor: Cornelius Schneider

Transaction Report:

Dear Prof. Kurumizaka,

Thank you for submitting your manuscript for consideration by the EMBO Journal. It has now been seen by three referees whose comments are shown below.

Given the referees' positive recommendations, I would like to invite you to submit a revised version of the manuscript, addressing the comments of all three reviewers. I should add that it is EMBO Journal policy to allow only a single round of revision, and acceptance of your manuscript will therefore depend on the completeness of your responses in this revised version.

Please contact me if you have any further questions. Thank you for the opportunity to consider your work for publication. I look forward to your revision.

Yours sincerely,

Cornelius Schneider

Cornelius Schneider, PhD
Editor
The EMBO Journal
c.schneider@embojournal.org

- a Reagents and Tools Table as part of the Methods section, which can be downloaded from our author guidelines

We realize that it is difficult to revise to a specific deadline. In the interest of protecting the conceptual advance provided by the work, we recommend a revision within 3 months (2nd Feb 2025). Please discuss the revision progress ahead of this time with the editor if you require more time to complete the revisions. Use the link below to submit your revision:

Referee #1:

Akatsu et al study RNA polymerase II transcription on a non-canonical nucleosome containing histone variant H2A.B (H2A.B nucleosome hereafter). They performed in vitro transcription experiments and showed that the transcription efficiency of RNA polymerase II elongation complex (RNAPII-EC) is higher on H2A.B nucleosomes than on the canonical H2A nucleosomes. They determined two cryo-EM structures of RNAPII-EC in complex with H2A.B nucleosomes, one with RNAPII-EC paused near the entry of nucleosome (at SHL-5), and the other one with RNAPII-EC stalled paused before reaching nucleosome dyad (at SHL-1). The authors found that the histone octamer remain intact when RNAPII-EC is paused at SHL-5, while the H2A.B-H2B dimer proximal to RNAPII is dissociated from the nucleosome when RNAPII-EC is paused at SHL-1. Such dissociation of the H2A-H2B dimer was not observed in previous structural study on the canonical nucleosomes. The authors proposed that the C-terminal region of H2A.B is responsible for the enhanced nucleosome transcription by RNAPII-EC. They generated a chimeric histone by stitching the C-terminal tail of H2A.B to H2A, and showed that the transcription efficiency on the nucleosome with the chimeric histone is increased and becomes comparable to H2A.B nucleosome.

In general, this study shows interesting findings of transcription on a non-canonical nucleosome with histone variant. The biochemical studies are solid. However, the cryo-EM density shown in this manuscript is not clear enough to support their findings (explained in major points). This manuscript can be published on EMBOJ if the following questions are addressed.

Major points:

1. The cryo-EM density of the H2A.B nucleosome of both structures, especially the density around histone H2A.B-H2B dimers, is key to the findings of this study. The reported resolution of the nucleosomes of both structures are relatively low (7.3 Å for structure paused at SHL-5 and 6.4 Å for the structure paused at SHL-1), which is at the boundary to tell whether a structural finding is reliable or not. I cannot find a focused cryo-EM density around the nucleosome of either structure. The snapshots of the nucleosome density in EV3C and EV5C only show one side of nucleosome, and I cannot see whether a H2A.B-H2B dimer is missing or not. To support their finding that a H2A.B-H2B dimer is dissociated when RNAP-EC is paused at SHL-1, the authors should show density maps of nucleosome from both structures, and clearly indicate where the dimer density is missing.
2. Related to the point above, the authors should also provide biochemical evidence to show that when RNAPII is stalled at SHL-1, one H2A.B-H2B dimer is dissociated.
3. In Fig3, the authors show that replacing the C-terminal region of H2A by that of H2A.B increases the efficiency of transcription. To corroborate the point that the C terminal region of H2A.B is responsible for the enhanced transcription, can the authors generate a nucleosome with chimeric H2A.B with H2A tail, e.g. H2A.BH2A (98-129), and perform the transcription assay?

Minor points:

1. Line 36: Claiming that "nucleosome has an extremely stable architecture ..." is too strong. Nucleosome it is a stable structure, but the DNA wrapped around it has 'breathing' effects and the position of nucleosome can be easily changed by either heating up or ATP-driven chromatin remodelers. It's better to put it as "nucleosome has a stable architecture ...".
2. The presentation of structures in Fig2 is hard to interpret of general audience. The authors show the overall structure paused at SHL-5 as a density map (panel B), but show the overall structure paused at SHL-1 as cartoon (panel D), which is hard for visual comparison. Panel C shows the RNAPII-EC paused at SHL-5. However, the RNAPII looks far away from SHL-5. I suggest the authors clarify how they define 'RNAPII paused at SHL-5 and SHL-1', and label key position like SHL0 (dyad), SHL-1 and SHL-5 on the structure.
3. Line 180: The authors discuss docking domain for several times, which it is important for the audience to understand the how a shorter H2A.B C-terminal region influence the wrapping of DNA. I suggest the authors indicate the docking domain in the figure (e.g. Fig. 3B).

Referee #2:

This manuscript is one in a series of RNA PolII /nucleosome cryo-EM studies from the same lab. In this particular instance, they focus on NCPs with H2A.B. They look at transcription and see less pausing at the SHL-5 position, accompanied by more efficient transcription. Previously it was already found (Commun Biol 7:1144 (2024)) that the H2A.B is a weaker binder. Here, cryo-EM analysis shows that the PolII interaction results in the loss of one of the H2A.B/H2b dimers from the nucleosome in the SHL-5 state. They then exchange the C-terminal peptide of H2A with that of H2A.B and see that this also destabilizes H2A, pointing to the tail as the region that mediates destabilization.

The work is competently executed, the gels are very nice, the structures bring the message, despite their extreme orientation problem, and the points are well made. Since the analyses are analogous to earlier papers, on a new histone variant, the interest is probably limited to researchers interested in H2A variants.

Referee #3:

In this study, the Kurumizaka lab investigates a novel mechanism through which the histone variant H2A.B, known to accumulate in actively transcribing genes, enhances transcriptional elongation. The authors demonstrate that RNA polymerase II's elongation complex (EC) transcribes through nucleosomes containing H2A.B more efficiently than through those with canonical H2A. To explore how the EC navigates DNA wrapped around H2A.B nucleosomes, the authors used cryo-EM to capture snapshots of EC-nucleosome complexes at various stages of RNAPII progression.

The structural analysis reveals that, when the EC reaches the SHL(-1) position on the H2A.B nucleosome, the proximal H2A.B-H2B dimer dissociates from the nucleosome. By contrast, in the canonical H2A nucleosome, the proximal H2A-H2B dimer remains intact when the EC pauses at the same SHL(-1) position. Interestingly, the authors observe that the H2A.B-H2B dimer remains bound to the nucleosome until the EC reaches the SHL(-5) position, suggesting that dimer dissociation is likely triggered after the EC transcribes beyond this point. The dissociation of the H2A.B-H2B dimer during transcription likely reduces the likelihood of nucleosome reassembly, potentially leading to nucleosome disassembly following EC passage. The study also presents intriguing links between this process and DNA damage repair pathways.

This is a significant and timely study that provides valuable new insights into the poorly understood function of the histone variant H2A.B. The experiments are well-conceived and executed, and the data are of high quality, clearly explained, and thoroughly discussed. For these reasons, I strongly recommend this manuscript for publication.

Regarding the final resolution of the cryo-EM map: while I do not have any technical concerns, I note that the resolution of the post-processed map for the Spt4/5-Elf1-RNAPII-H2A.B nucleosome, paused at the SHL(-5) position, is somewhat low in the nucleosome region. It would be helpful if the authors could briefly clarify why this does not affect their conclusions for the sake of clarity.

Referee Cross-commenting:

Editor:

Dear referees,

thank you very much for your help and your informative input. Please find the comments of the other referees below. I would be very thankful if you could give additional input here.

As you can see referee #2 does not think that there is sufficient novelty compared to the recent publication by the same authors (PMID: 39277674). Given the more positive assessment by referees #1 and #3 I would in principle be open to consider this manuscript further nonetheless. However, I am concerned that the mechanistic and molecular insight might also be limited given that the structures appear to be very low resolution in regions that are important to provide detailed insight into the proposed H2A.B-H2B dimer eviction mechanism. Referee #1 even remarks that it is never explicitly visible in the structures that H2A.B-H2B is released. Taken together I have the impression now that the advance is indeed too limited in the current iteration of the manuscript. Given the overall positive evaluation by referees #1 and #3 I am wondering if I am underestimating the finding. Do you think that there is sufficient support that PolII passage through the nucleosome is indeed causing a release of H2A.B-H2B? Thank you for your help with this manuscript.

With best regards,
Cornelius

Ref #1

Dear editor,

Thank you very much for raising the discussion. In my opinion, the previous publication from by the same authors (PMID: 39277674) does not downplay the novelty of this manuscript. In their previous publication, the authors found that the H2A.B nucleosomes unwind DNA at a faster rate, and the H2A.B-H2B dimer is easier to dissociate from DNA, compared to the canonical nucleosomes containing H2A. That study focuses on the dynamics of the nucleosome itself by applying nanopore measurements and MD simulations. In comparison, the current manuscript focuses on the interplay between RNA polymerase II elongation complex and the H2A.B containing nucleosomes, which shows that the more dynamic H2A.B nucleosomes do promote transcription, and provides structural explanation. However, the cryoEM results they show in the current manuscript raise my concern. The authors should provide better cryoEM maps to support their conclusions. I suggest they also provide corresponding cryoEM map files to reviewers if there is further revision.

Ref #3

Dear all,

I agree with Referee #1 that there is substantial novelty. I do not have concerns regarding the cryo-EM data, but it would be good if the authors could discuss the question of resolution and better explain why their cryo-EM maps support their conclusions.

Ref #2

Dear all,

The structure is low resolution indeed, but should be sufficient to give the current conclusion, although side-by-side comparison is indeed needed, as pointed out by reviewer 1. It is not necessarily clear that higher resolution data would provide additional mechanistic insight, which is why we did not ask for that. But indeed, the transcription gels are nice and of good quality, so if other reviewers like it very much I have no problem with a revised version.

Referee #1:

Akatsu et al study RNA polymerase II transcription on a non-canonical nucleosome containing histone variant H2A.B (H2A.B nucleosome hereafter). They performed in vitro transcription experiments and showed that the transcription efficiency of RNA polymerase II elongation complex (RNAPII-EC) is higher on H2A.B nucleosomes than on the canonical H2A nucleosomes. They determined two cryo-EM structures of RNAPII-EC in complex with H2A.B nucleosomes, one with RNAPII-EC paused near the entry of nucleosome (at SHL-5), and the other one with RNAPII-EC stalled paused before reaching nucleosome dyad (at SHL-1). The authors found that the histone octamer remain intact when RNAPII-EC is paused at SHL-5, while the H2A.B-H2B dimer proximal to RNAPII is dissociated from the nucleosome when RNAPII-EC is paused at SHL-1. Such dissociation of the H2A-H2B dimer was not observed in previous structural study on the canonical nucleosomes. The authors proposed that the C-terminal region of H2A.B is responsible for the enhanced nucleosome transcription by RNAPII-EC. They generated a chimeric histone by stitching the C-terminal tail of H2A.B to H2A, and showed that the transcription efficiency on the nucleosome with the chimeric histone is increased and becomes comparable to H2A.B nucleosome.

In general, this study shows interesting findings of transcription on a non-canonical nucleosome with histone variant. The biochemical studies are solid. However, the cryo-EM density shown in this manuscript is not clear enough to support their findings (explained in major points). This manuscript can be published on EMBOJ if the following questions are addressed.

Reply)

Thank you very much for your favorable comments. We revised the manuscript according to this reviewer's comments, as described below.

Major points:

1. The cryo-EM density of the H2A.B nucleosome of both structures, especially the density around histone H2A.B-H2B dimers, is key to the findings of this study. The reported resolution of the nucleosomes of both structures are relatively low (7.3 Å for structure paused at SHL-5 and 6.4 Å for the structure paused at SHL-1), which is at the boundary to tell whether a structural finding is reliable or not. I

cannot find a focused cryo-EM density around the nucleosome of either structure. The snapshots of the nucleosome density in EV3C and EV5C only show one side of nucleosome, and I cannot see whether a H2A.B-H2B dimer is missing or not. To support their finding that a H2A.B-H2B dimer is dissociated when RNAP-EC is paused at SHL-1, the authors should show density maps of nucleosome from both structures, and clearly indicate where the dimer density is missing.

Reply)

Thank you very much for this comment. According to this reviewer's suggestion, we re-analyzed the EC-nucleosome complexes, and now present new cryo-EM maps with higher resolutions. We then improved the nucleosome resolution in the SHL-5 complex from 7.3 Å to 6.9 Å and that in the SHL-1 complex from 6.4 Å to 4.8 Å. We described these results on p.6 l.112. In addition, we presented the cryo-EM maps of the nucleosomes in the complexes in the new Fig. 2C and Fig. EV5, according to this reviewer's suggestion.

2. Related to the point above, the authors should also provide biochemical evidence to show that when RNAPII is stalled at SHL-1, one H2A.B-H2B dimer is dissociated.

Reply)

Thank you very much for this insightful comment. To detect the H2A.B-H2B dissociation during transcription elongation, we performed the DNaseI footprinting analysis with the nucleosome sample, in which the transcribing RNAPII elongation complex was paused at the SHL-1 position. We then found that, in the H2A.B nucleosome, the nucleosomal DNA region near the proximal H2A.B-H2B dimer becomes accessible to DNaseI in a transcription elongation-dependent manner. These DNaseI sensitive sites were not observed in the canonical nucleosome with the EC paused at the SHL-1 position. These results are perfectly consistent with the cryo-EM structure, in which the proximal H2A.B-H2B dimer is disassembled when the transcribing EC reaches the SHL-1 position. These new data are presented in the new Fig. 3, Appendix Fig. S4, and described on p. 7, l. 134.

3. In Fig3, the authors show that replacing the C-terminal region of H2A by that of H2A.B increases the efficiency of transcription. To corroborate the point that

the C terminal region of H2A.B is responsible for the enhanced transcription, can the authors generate a nucleosome with chimeric H2A.B with H2A tail, e.g. H2A.BH2A (98-129), and perform the transcription assay?

Reply)

According to this reviewer's suggestion, we prepared a chimeric H2A.B with the H2A tail, H2A.B•H2A (98-129), and performed the transcription assay. We then found that the H2A.B•H2A (98-129) mutant exhibits a quite similar profile to that of H2A.B in nucleosome transcription. These new results are presented in the new Fig. 4C, Appendix Fig. S2C, and S2D, and described on p.9 l.173. The similar transcription profiles may be a consequence of the H2A.B-specific residues in the histone-fold domain, which were previously tested in the MNase susceptibility assay from Dr. Zhou's Lab (Zhou et al, EMBO J., 2020). We also discuss this fact in a new paragraph of the Discussion section (p. 11).

Minor points:

1. Line 36: Claiming that "nucleosome has an extremely stable architecture ..." is too strong. Nucleosome it is a stable structure, but the DNA wrapped around it has 'breathing' effects and the position of nucleosome can be easily changed by either heating up or ATP-driven chromatin remodelers. It's better to put it as "nucleosome has a stable architecture ...".

Reply)

We corrected it accordingly.

2. The presentation of structures in Fig2 is hard to interpret of general audience. The authors show the overall structure paused at SHL-5 as a density map (panel B), but show the overall structure paused at SHL-1 as cartoon (panel D), which is hard for visual comparison. Panel C shows the RNAPII-EC paused at SHL-5. However, the RNAPII looks far away from SHL-5. I suggest the authors clarify how they define 'RNAPII paused at SHL-5 and SHL-1', and label key position like SHL0 (dyad), SHL-1 and SHL-5 on the structure.

Reply)

Thank you very much. We defined the RNAPII pausing positions by the RNA sequence in the RNAPII catalytic center for the SHL(-1) complex. This was

possible because the RNA nucleotide incorporated in the RNAPII catalytic center are clearly visible in the cryo-EM structures, as mentioned on p.6. For the SHL(-5) complex, the resolution of the structure was not sufficient to evaluate the pausing position. We then deduced its pausing position from the length of the RNA transcripts. This was mentioned on p.6 l.112. In addition, according to this reviewer's suggestion, we substantially improved Fig. 2 by presenting new structures with better resolutions.

3. Line 180: The authors discuss docking domain for several times, which it is important for the audience to understand the how a shorter H2A.B C-terminal region influence the wrapping of DNA. I suggest the authors indicate the docking domain in the figure (e.g. Fig. 3B).

Reply)

We indicated the amino acid residues in the docking domain in the new Fig. 4A and added the structural figures for the docking domain as the new Fig. 4B.

Referee #2:

This manuscript is one in a series of RNA PolII /nucleosome cryo-EM studies from the same lab. In this particular instance, they focus on NCPs with H2A.B. They look at transcription and see less pausing at the SHL-5 position, accompanied by more efficient transcription. Previously it was already found (Commun Biol 7:1144 (2024)) that the H2A.B is a waker binder. Here, cryo-EM analysis shows that the PolII interaction results in the loss of one of the H2A.B/H2b dimers from the nucleosome in the SHL-5 state. They then exchange the C-terminal peptide of H2A with that of H2A.B and see that this also destabilizes H2A, pointing to the tail as the region that mediates destabilization.

The work is competently executed, the gels are very nice, the structures bring the message, despite their extreme orientation problem, and the points are well made. Since the analyses are analogous to earlier papers, on a new histone variant, the interest is probably limited to researchers interested in H2A variants.

Reply)

Thank you very much for your thoughtful comments. Histone variants play crucial roles in epigenetic genome regulation, yet their specific behaviors in transcription remain largely unexplored. Given that H2A.B is associated with transcriptionally active genomic regions, understanding its role during elongation is an important issue to address.

In our revised manuscript, we have re-analyzed the cryo-EM structures of RNAPII EC-H2A.B nucleosome complexes and provided clearer cryo-EM maps, now presented in the updated Figure 2C and Figure EV5. To further investigate H2A.B-H2B eviction during transcription elongation, we conducted a DNaseI footprinting analysis, which revealed that in the H2A.B nucleosome, the DNA region near the proximal H2A.B-H2B dimer becomes more accessible to DNaseI in a transcription elongation-dependent manner. Notably, these DNaseI-sensitive sites were absent in the canonical nucleosome when the EC-nucleosome complex was paused at the SHL-1 position.

These structural and biochemical findings consistently suggest that in the H2A.B nucleosome, the proximal H2A.B-H2B dimer is disassembled when the transcribing EC reaches the SHL-1 position. These new data support our conclusions and broaden the study's relevance beyond H2A variant researchers to those interested in chromatin structure and transcription regulation.

We appreciate your valuable feedback, which has helped us refine and strengthen our study.

Referee #3:

In this study, the Kurumizaka lab investigates a novel mechanism through which the histone variant H2A.B, known to accumulate in actively transcribing genes, enhances transcriptional elongation. The authors demonstrate that RNA polymerase II's elongation complex (EC) transcribes through nucleosomes containing H2A.B more efficiently than through those with canonical H2A. To explore how the EC navigates DNA wrapped around H2A.B nucleosomes, the authors used cryo-EM to capture snapshots of EC-nucleosome complexes at various stages of RNAPII progression.

The structural analysis reveals that, when the EC reaches the SHL(-1) position on the H2A.B nucleosome, the proximal H2A.B-H2B dimer dissociates from the nucleosome. By contrast, in the canonical H2A nucleosome, the proximal H2A-H2B dimer remains intact when the EC pauses at the same SHL(-1) position. Interestingly, the authors observe that the H2A.B-H2B dimer remains bound to the nucleosome until the EC reaches the SHL(-5) position, suggesting that dimer dissociation is likely triggered after the EC transcribes beyond this point. The dissociation of the H2A.B-H2B dimer during transcription likely reduces the likelihood of nucleosome reassembly, potentially leading to nucleosome disassembly following EC passage. The study also presents intriguing links between this process and DNA damage repair pathways.

This is a significant and timely study that provides valuable new insights into the poorly understood function of the histone variant H2A.B. The experiments are well-conceived and executed, and the data are of high quality, clearly explained, and thoroughly discussed. For these reasons, I strongly recommend this manuscript for publication.

Regarding the final resolution of the cryo-EM map: while I do not have any technical concerns, I note that the resolution of the post-processed map for the Spt4/5-Elf1-RNAPII-H2A.B nucleosome, paused at the SHL(-5) position, is somewhat low in the nucleosome region. It would be helpful if the authors could briefly clarify why this does not affect their conclusions for the sake of clarity.

Reply)

Thank you very much for your favorable comments. We acknowledge that the resolution of the post-processed cryo-EM map for the Spt4/5-Elf1-RNAPII-H2A.B nucleosome, paused at the SHL(-5) position, is relatively lower in the nucleosome region compared to other parts of the complex. This is likely due to the intrinsic flexibility of the nucleosome in this state, which leads to local heterogeneity and reduced resolution in the reconstructions. We mentioned this fact in the revised manuscript (p.6 l.112). In addition, we re-analyzed the EC-nucleosome complexes, and now present cryo-EM maps with higher resolutions. We then improved the nucleosome resolution in the SHL-5 complex from 7.3 Å to 6.9 Å, and that in the SHL-1 complex from 6.4 Å to 4.8 Å. We described these results

on p.6 l.112, and presented the cryo-EM maps of the nucleosomes in the complexes in the new Fig. 2C and Fig. EV5.

Referee Cross-commenting:

Editor:

Dear referees,

thank you very much for your help and your informative input. Please find the comments of the other referees below. I would be very thankful if you could give additional input here.

As you can see referee #2 does not think that there is sufficient novelty compared to the recent publication by the same authors (PMID: 39277674). Given the more positive assessment by referees #1 and #3 I would in principle be open to consider this manuscript further nonetheless. However, I am concerned that the mechanistic and molecular insight might also be limited given that the structures appear to be very low resolution in regions that are important to provide detailed insight into the proposed H2A.B-H2B dimer eviction mechanism. Referee #1 even remarks that it is never explicitly visible in the structures that H2A.B-H2B is released. Taken together I have the impression now that the advance is indeed too limited in the current iteration of the manuscript. Given the overall positive evaluation by referees #1 and #3 I am wondering if I am underestimating the finding. Do you think that there is sufficient support that PolIII passage through the nucleosome is indeed causing a release of H2A.B-H2B?

Thank you for your help with this manuscript.

With best regards,

Cornelius

Ref #1

Dear editor,

Thank you very much for raising the discussion. In my opinion, the previous publication from by the same authors (PMID: 39277674) does not downplay the novelty of this manuscript. In their previous publication, the authors found that the

H2A.B nucleosomes unwind DNA at a faster rate, and the H2A.B-H2B dimer is easier to dissociate from DNA, compared to the canonical nucleosomes containing H2A. That study focuses on the dynamics of the nucleosome itself by applying nanopore measurements and MD simulations. In comparison, the current manuscript focuses on the interplay between RNA polymerase II elongation complex and the H2A.B containing nucleosomes, which shows that the more dynamic H2A.B nucleosomes do promote transcription, and provides structural explanation. However, the cryoEM results they show in the current manuscript raise my concern. The authors should provide better cryoEM maps to support their conclusions. I suggest they also provide corresponding cryoEM map files to reviewers if there is further revision.

Reply)

Thank you very much. We re-analyzed the cryo-EM structures, and the cryo-EM maps overlapped with the nucleosome models are now presented in the new Fig. 2C. We also provided cryo-EM maps of the H2A.B nucleosomes in the transcribing RNAPII EC-nucleosome complexes paused at the SHL(-5) and SHL(-1) positions with different contour levels in Fig. EV5. These new data now clearly show the absence of the proximal H2A.B-H2B dimer in the SHL(-1) complex, but not in the SHL(-5) complex.

Ref #3

Dear all,

I agree with Referee #1 that there is substantial novelty. I do not have concerns regarding the cryo-EM data, but it would be good if the authors could discuss the question of resolution and better explain why their cryo-EM maps support their conclusions.

Reply)

Thank you very much. As described above, we re-analyzed the cryo-EM structures and provided refined cryo-EM maps in the revised Fig. 2C and Fig. EV5.

Ref #2

Dear all,

The structure is low resolution indeed, but should be sufficient to give the current conclusion, although side-by-side comparison is indeed needed, as pointed out by reviewer 1. It is not necessarily clear that higher resolution data would provide additional mechanistic insight, which is why we did not ask for that. But indeed, the transcription gels are nice and of good quality, so if other reviewers like it very much I have no problem with a revised version.

Reply)

Thank you very much. In the revised manuscript, we re-analyzed the cryo-EM structures, and provided refined cryo-EM maps in the revised Fig. 2C and Fig. EV5. In addition, we performed the DNaseI footprinting analysis, and obtained new results that, in the H2A.B nucleosome, the nucleosomal DNA region near the proximal H2A.B-H2B dimer becomes accessible to DNaseI in a transcription elongation-dependent manner. These DNaseI sensitive sites were not observed in the canonical nucleosome with the EC-nucleosome complex paused at the SHL-1 position. These results suggest that, in the H2A.B nucleosome, the proximal H2A.B-H2B dimer is disassembled when the transcribing EC reaches the SHL-1 position. These new cryo-EM and *in vitro* transcription results consistently showed that the proximal H2A.B-H2B dimer is disassembled while the transcribing RNAPII is traversing the proximal half of the nucleosomal DNA.

Dear Dr Kurumizaka,

Thank you for submitting a revised version of your manuscript. Your study has now been seen by all original referees, who find that their previous concerns have been addressed and now recommend publication of the manuscript. There remain only a few mainly editorial points that have to be addressed before I can extend formal acceptance of the manuscript:

- Please double-check to make sure to all relevant funding information in the manuscript is also entered into our submission system. (Missing in the system currently: JSPS KAKENHI Grant Numbers JP22K15033)
- On the abstract page of the manuscript, please include 4-5 general keyword terms to enhance searchability.
- Please adjust the format of the reference list and of the in-text citations according to EMBO Journal format (alphabetical order, author name et al + year.../up to 10 author names in the reference list before et al / please refer to our Guide to Authors for additional information on EMBO J reference format).
- Please rename the Conflict-of-Interest section into "Disclosure and Competing Interests Statement", in accordance with our updated Guide to Authors (<https://www.embopress.org/competing-interests>)
- As we are switching from a free-text author contribution statement towards a more formal statement based on Contributor Role Taxonomy (CRediT) terms, please remove the present Author Contribution section and instead specify each author's contribution(s) directly in the Author Information page of our submission system during upload of the final manuscript. See <https://casrai.org/credit/> for more information.
- Please list all figure callouts sequentially; Please also note that there is a missing callouts for Appendix Fig. S3
- Please save the Source data files need to be saved in a scheme one figure/folder and then uploaded as .zip files. E.g. all the Source data files for figure 1 need to be saved in a single folder and this needs to be zipped and then uploaded as "SD figure 1.zip" file. For EV and/or appendix figures, ZIP together all source data. Completed SD checklist should be uploaded as Related Manuscript File.
- Please provide the specific URLs for EMD-60592, EMD-60593, 9II7 datasets in the data availability statement.
- Figure Legends (main + EV): Please note that the measure of center for the error bars needs to be defined in the legends of figures 1D, 4D.
- Please remove the "Conventions and Abbreviations" section from the manuscript
- Section order should be corrected: Title page - Abstract & Keywords - Introduction - Results - Discussion - Methods - Data Availability - Acknowledgements - Disclosure and Competing Interests Statement - References - Figure Legends - Table(s) - Expanded View Figure Legends.

With best regards,

Cornelius Schneider

Cornelius Schneider, PhD
Editor | The EMBO Journal
c.schneider@embojournal.org

Use the link below to submit your revision:

Referee #1:

The authors have appropriately addressed all of my comments and concerns.

All editorial and formatting issues were resolved by the authors.

Dear Prof. Kurumizaka,

I am pleased to inform you that your manuscript has been accepted for publication in the EMBO Journal.

Yours sincerely,

Cornelius Schneider, PhD
Editor
The EMBO Journal
c.schneider@embojournal.org
